

# Multi-orbital two-particle self-consistent approach – Strengths and limitations

Jonas B. Profe[1][⋆], Jiawei Yan[2], Karim Zantout[1], Philipp Werner[2] and Roser Valentí[1]

**1** Institute for Theoretical Physics, Goethe University Frankfurt,
Max-von-Laue-Straße 1, D-60438 Frankfurt a.M., Germany
**2** Department of Physics, University of Fribourg, 1700 Fribourg, Switzerland

⋆ profe@itp.uni-frankfurt.de

## Abstract

Extending many-body numerical techniques which are powerful in the context of simple model calculations to the realm of realistic material simulations can be a challenging task. Realistic systems often involve multiple active orbitals, which increases the complexity and numerical cost because of the large local Hilbert space and the large number of interaction terms or sign-changing off-diagonal Green's functions. The two-particle self-consistent approach (TPSC) is one such many-body numerical technique, for which multi-orbital extensions have proven to be involved due to the substantially more complex structure of the local interaction tensor. In this paper we extend earlier multi-orbital generalizations of TPSC by setting up two different variants of a fully self-consistent theory for TPSC in multi-orbital systems. We first investigate the strengths and limitations of the approach analytically and then benchmark both variants against dynamical mean-field theory (DMFT) and D-TRILEX results. We find that the exact behavior of the system can be faithfully reproduced in the weak-coupling regime, while at stronger couplings the performance of the two TPSC variants strongly depends on details of the system.

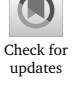

# 1  Introduction

Solving the many-body problem of interacting electrons remains a central challenge in condensed matter physics. Analytical solutions can only be found in rare cases [1,2], necessitating the development of advanced numerical methods such as Density Functional Theory [3, 4], Quantum Monte Carlo (QMC) [5–10], DMRG and other Tensor network approaches [11–13], Dynamical Mean-Field Theory and its extensions [14–19], variational (neural) quantum states [10,20–25] or various diagrammatic methods [26–31]. With the use of these approaches the community gained insights into correlated electron physics, e.g. by developing an understanding of Mott physics [32–35], (unconventional) superconductivity [36–40], the pseudogap phase [41,42], magnetism, spin liquids [43–45] and charge ordered states [46–48]. In recent years, a coherent picture has emerged for some single-orbital models, with a variety of numerical approaches producing consistent results for ground state energies, mass renormalizations, etc. [49–51], which shifts the frontiers of method development to more complex models and setups.

Going beyond single-orbital models, it is natural to consider multi-orbital extensions which are relevant for real materials. In such multi-orbital models, the complexity of the orbital structure introduces new competing energy scales, as exemplified by the Hund's metals [52–56] which turn out to be relevant for a large variety of materials including iron-based superconductors [54, 57–60], ruthenates [61, 62] and molybdates [63], to mention a few. This more intricate local structure of the interaction tensor has profound consequences for a number of numerical approaches, e.g. leading to a sign problem in some QMC variants [64] and, in general, to a larger numerical cost. Thus, extending numerical techniques to multi-orbital systems poses often not only an analytical but also a computational challenge which has to be overcome. Furthermore, approximations known to work well in the single-orbital case are not guaranteed to be equally adequate for multi-orbital systems.

Numerical studies can provide guiding principles and insights into the microscopic mechanisms behind emergent phenomena. However, in cases where the employed numerical approach fails qualitatively or quantitatively, as for example in the case of some iron-based superconductors where neither DFT nor DFT+DMFT predict correct Fermi surfaces [65–67], it is important to extend the techniques and improve their accuracy. A promising route for the study of correlated materials is to extend the formalism beyond DMFT, which is a dynami-

cal but local approach, by combining it with another numerical approach which captures the effects of spatial and temporal fluctuations [14]. For this, two different strategies have been considered in recent years. One idea is to include finite distance correlations in the DMFT itself, at the cost of a more complicated impurity problem, as in the case of cellular DMFT [68–70] and the dynamical cluster approximation [71,72]. Alternatively, one can extend the locality-approximation of DMFT from the self-energy to some vertex-functions, as done for example in TRILEX [18] and D-TRILEX [73,74], the dynamical-vertex approximation [17,75] and other schemes [14,76–78]. These approaches however typically require the calculation of local vertex functions, which is a time consuming and challenging task for complicated models [79].

One very successful numerical method for systems with weak to intermediate correlations is the two particle self-consistent (TPSC) approach [80–82], which was first formulated for the single-band Hubbard model [83,84], and subsequently extended to multi-site [85,86], non-$SU(2)$ [87,88], multi-orbital [66,89–91] and non-equilibrium [92,93] problems. Furthermore, it has been combined with DMFT to extend its range of validity [94–96]. TPSC has been applied extensively to single-orbital models [80,97–104], for which it yields remarkably accurate results [50] at a comparably low numerical cost. Its combination with DMFT does not require the calculation of a vertex, but only the two-particle density matrix, which can be evaluated at much lower numerical cost. If this methodology works reliably in multi-orbital systems, TPSC and TPSC+DMFT will become prime contenders for the study of complex correlated materials.

Motivated by these prospects, we present and benchmark in this paper two different variants of a fully self-consistent multi-orbital TPSC approach by establishing sum rules taking into account the $SU(2)$ symmetry of the system. This allows us to determine all required two-particle expectation values (TPEV) from exact sum rules, overcoming approximations that had to be applied in earlier formulations [89,90].

The paper is structured as follows: In Section 2, we derive the central equations of multi-orbital TPSC from scratch. In Section 3.1, we analyze a density-density interaction-only model analytically within TPSC, and point out potential pitfalls. In Section 3.2 we analyze the effects of a strong Hunds coupling. Next, in Section 3.3, we benchmark the accuracy of the self-consistently determined two-particle expectation values, by comparing the TPSC results to DMFT calculations. In addition, we compare the self-energy, spin and charge susceptibilities to D-TRILEX [19] and discuss the implications of these results for the applicability of our TPSC formulations to realistic multi-orbitals models.

## 2 Multi-orbital TPSC

In this chapter we present two variants of the multi-orbital TPSC formalism. The derivation follows closely analogous derivations for the single-orbital case [80,82]. In the following we will consider a general tight-binding Hamiltonian with local and instantaneous interactions

$$
\begin{aligned}
H = &\, t_{o_{1'},o_{3'}}(\mathbf{r}'_1 - \mathbf{r}'_3) c^{\dagger}_{o_{3'},s'}(\mathbf{r}'_3) c_{o_{1'},s'}(\mathbf{r}'_1) \\
&+ \frac{1}{2} \tilde{V}^{s_{1'}s_{2'}s_{3'}s_{4'}}_{o_{1'}o_{2'}o_{3'}o_{4'}} c^{\dagger}_{o_{3'},s_{3'}}(\mathbf{r}') c^{\dagger}_{o_{4'},s_{4'}}(\mathbf{r}') c_{o_{2'},s_{2'}}(\mathbf{r}') c_{o_{1'},s_{1'}}(\mathbf{r}'),
\end{aligned}
\tag{1}
$$

where $c^{(\dagger)}_{o_1,s_1}(\mathbf{r}_1)$ annihilates (creates) an electron at orbital $o_1$ with spin $s_1$ at the lattice position $\mathbf{r}_1$. $t_{o_1,o_3}(\mathbf{r}_1 - \mathbf{r}_3)$ is the hopping matrix element between two orbitals at distance $\mathbf{r}_1 - \mathbf{r}_3$ and $\tilde{V}^{s_1 s_2 s_3 s_4}_{o_1 o_2 o_3 o_4}$ is the local interaction matrix element. All primed variables are summed over. For the rest of the paper we will work in the Baym-Kadanoff formalism, thus the operators $c_{o_1,s_1}(\mathbf{r})$ acquire a dependence on the imaginairy time $c_{o_1,s_1}(\mathbf{r}, \tau)$. We introduce the four vector



$\tau \equiv (r, \tau)$ as a short hand to compactify our notation. Furthermore, we define the anti-symmetrized interaction tensor

$$V^{s_1 s_2 s_3 s_4}_{o_1 o_2 o_3 o_4} = \frac{1}{2} \mathcal{P} \left( \tilde{V}^{s_1 s_2 s_3 s_4}_{o_1 o_2 o_3 o_4} \right), \tag{2}$$

where $\mathcal{P}$ is the anti-symmetrization operator. From now on, we will work with the antisymmetrized interation. In App. B we perform the analogous derivations starting from a Hubbard-Kanamori type instead of a general anti-symmetrical interaction tensor, which leads to an equivalent set of self-consistency equations. We will restrict ourselves to the case of an $SU(2)$ symmetric model. To reduce the complexity, we rewrite the interaction tensor in terms of its even and odd $SU(2)$-transforming components [105]

$$V^{s_1 s_2 s_3 s_4}_{o_1 o_2 o_3 o_4} = U_{o_1 o_2 o_3 o_4} \delta_{s_1, s_3} \delta_{s_2, s_4} - U_{o_1 o_2 o_4 o_3} \delta_{s_1, s_4} \delta_{s_2, s_3}. \tag{3}$$

Lastly, we restrict ourselves to inter-orbital-bilinear type interactions [106] (still allowing for intra-orbital interactions within this bilinear form), thus the spin independent interaction tensor simplifies to

$$U_{o_1 o_2 o_3 o_4} = D_{o_1, o_4} \delta_{o_1, o_3} \delta_{o_2, o_4} + C_{o_1, o_3} \delta_{o_1, o_4} \delta_{o_2, o_3} + P_{o_1, o_3} \delta_{o_1, o_2} \delta_{o_3, o_4}, \tag{4}$$

where each of the contributions is native to a different diagram type: $D$ has the form of a resummation in the direct particle-hole channel, $C$ has the form of a crossed particle-hole one and a particle-particle resummation leads to terms of the form of $P$. TPSC can also be formulated without this restriction, however restricting the interaction allows for an easier understanding of the equations and processes involved.

The central equation on which TPSC is built is the equation of motion [80], linking the product of the single particle self-energy and the Green's function to the tensor contraction of the two-particle interaction with a two-particle expectation value (TPEV). The TPEV can then be re-expressed in terms of a generalized susceptibility. For a derivation of the equation of motion, see Appendix A up to Eq. (A.19). With the assumption of an $SU(2)$-symmetric inter-orbital bilinear-type interaction, the equation of motion simplifies to

$$\Sigma^{s_1, s_{1'}}_{o_1, o_{1'}}(\tau_1, \tau'_1) G^{s_{1'}, s_3}_{o_{1'}, o_3}(\tau'_1, \tau_3) = \frac{1}{2} V^{s_{4'}, s_{2'}, s_{3'}, s_1}_{o_{4'}, o_{2'}, o_{3'}, o_1} \langle T_\tau c^\dagger_{o_{3'}, s_{3'}}(\tau_1) c_{o_{2'}, s_{2'}}(\tau_1) c_{o_{4'}, s_{4'}}(\tau_1) c^\dagger_{o_3, s_3}(\tau_3) \rangle \tag{5}$$

$$= \frac{1}{2} U_{o_{4'}, o_{2'}, o_{3'}, o_1} \langle T_\tau c^\dagger_{o_{3'}, s_{4'}}(\tau_1) c_{o_{2'}, s_1}(\tau_1) c_{o_{4'}, s_{4'}}(\tau_1) c^\dagger_{o_3, s_3}(\tau_3) \rangle \tag{6}$$

$$\quad - \frac{1}{2} U_{o_{4'}, o_{2'}, o_1, o_{3'}} \langle T_\tau c^\dagger_{o_{3'}, s_{4'}}(\tau_1) c_{o_{2'}, s_{4'}}(\tau_1) c_{o_{4'}, s_1}(\tau_1) c^\dagger_{o_3, s_3}(\tau_3) \rangle$$

$$= U_{o_{4'}, o_{2'}, o_{3'}, o_1} \langle T_\tau c^\dagger_{o_{3'}, s_{4'}}(\tau_1) c_{o_{2'}, s_1}(\tau_1) c_{o_{4'}, s_{4'}}(\tau_1) c^\dagger_{o_3, s_3}(\tau_3) \rangle \tag{7}$$

$$= \Big( D_{o_{4'}, o_1} \langle T_\tau c^\dagger_{o_{4'}, s_{4'}}(\tau_1) c_{o_1, s_1}(\tau_1) c_{o_{4'}, s_{4'}}(\tau_1) c^\dagger_{o_3, s_3}(\tau_3) \rangle \tag{8}$$

$$\quad + C_{o_1, o_{3'}} \langle T_\tau c^\dagger_{o_{3'}, s_{4'}}(\tau_1) c_{o_{3'} s_1}(\tau_1) c_{o_1, s_{4'}}(\tau_1) c^\dagger_{o_3, s_3}(\tau_3) \rangle$$

$$\quad + P_{o_{2'}, o_1} \langle T_\tau c^\dagger_{o_1, s_{4'}}(\tau_1) c_{o_{2'} s_1}(\tau_1) c_{o_{2'}, s_{4'}}(\tau_1) c^\dagger_{o_3, s_3}(\tau_3) \rangle \Big).$$

The factor of 2 canceling the 1/2 stems from applying the remaining crossing symmetry (exchanging in-going and out-going indices at the same time), then swapping two operators and renaming summation indices. Furthermore, in the $P$ channel the only contribution to the spin sum which is non-zero is $s_1 \neq s'_4$ due to the Pauli-principle.

The equation of motion relates single-particle to two-particle properties, which in turn are linked to three-particle properties. Therefore, the set of equations is not amenable to an exact solution due to this hierarchical structure of the equations. Instead, we proceed by

approximating the two-particle expectation values on the right-hand side by their Hartree-Fock decoupling. However, to improve over Hartree-Fock, in TPSC one introduces parameters for the prefactors which are chosen such that the local and static limit of the two-particle expectation values fulfill exact local susceptibility sumrules [80]. In the multi-orbital case this means that we consider the limit $\tau_1 = \tau_3$, $o_1 = o_3$ and $s_1 = s_3$. Technically, the choice of the Ansätze is completely arbitrary, here we restrict ourselves to the cases in which the different interaction terms do not mix. The Hartree-Fock decoupling with the Ansätze introduced then reads

$$
\begin{aligned}
&\Sigma^{s_1,s_{1'}}_{o_1,o_{1'}}(\tau_1;\tau_1')G^{s_{1'},s_3}_{o_{1'},o_3}(\tau_1';\tau_3)\\
&\approx \Big(\tilde{D}^{s_{4'},s_1}_{o_{4'},o_1}\Big(G^{s_{4'},s_{4'}}_{o_{4'},o_{4'}}(\tau_1;\tau_1)G^{s_1,s_3}_{o_1,o_3}(\tau_1;\tau_3)-G^{s_1,s_{4'}}_{o_1,o_{4'}}(\tau_1;\tau_1)G^{s_{4'},s_3}_{o_{4'},o_3}(\tau_1;\tau_3)\Big)\\
&\quad + \tilde{C}^{s_1,s_{3'}}_{o_1,o_{3'}}\Big(G^{s_{4'},s_{4'}}_{o_1,o_{3'}}(\tau_1;\tau_1)G^{s_1,s_3}_{o_{3'},o_3}(\tau_1;\tau_3)-G^{s_1,s_{4'}}_{o_{3'},o_{3'}}(\tau_1;\tau_1)G^{s_{4'},s_3}_{o_1,o_3}(\tau_1;\tau_3)\Big)\\
&\quad + \tilde{P}^{s_1,s_1}_{o_{2'},o_1}\Big(G^{\bar{s}_1,\bar{s}_1}_{o_{2'},o_1}(\tau_1;\tau_1)G^{s_1,s_3}_{o_{2'},o_3}(\tau_1;\tau_3)-G^{s_1,\bar{s}_1}_{o_{2'},o_1}(\tau_1;\tau_1)G^{\bar{s}_1,s_3}_{o_{2'},o_3}(\tau_1;\tau_3)\Big)\Big),
\end{aligned}
\tag{9}
$$

where $\bar{s} = -s$. One important thing to keep in mind is that TPSC will partially inherit shortcomings of this underlying Hartree-Fock decoupling (we will discuss this in more detail in Sec 3.1). On the other hand, from Eq. (8) we find in the local and static limit

$$
\begin{aligned}
\Sigma^{s_1,s_{1'}}_{o_1,o_{1'}}(\tau_1,\tau_1')G^{s_{1'},s_1}_{o_{1'},o_1}(\tau_1',\tau_1)=\Big(&D_{o_{4'},o_1}\langle T_\tau c^\dagger_{o_{4'},s_{4'}}(\tau_1)c_{o_{4'},s_{4'}}(\tau_1)c^\dagger_{o_1,s_1}(\tau_1)c_{o_1,s_1}(\tau_1)\rangle\\
&- C_{o_1,o_{3'}}\langle T_\tau c^\dagger_{o_{3'},s_{4'}}(\tau_1)c_{o_{3'}s_1}(\tau_1)c^\dagger_{o_1,s_1}(\tau_1)c_{o_1,s_{4'}}(\tau_1)\rangle\\
&- P_{o_{2'},o_1}\langle T_\tau c^\dagger_{o_1,\bar{s}_1}(\tau_1)c_{o_{2'}s_1}(\tau_1)c^\dagger_{o_1,s_1}(\tau_1)c_{o_{2'},\bar{s}_1}(\tau_1)\rangle\Big).
\end{aligned}
\tag{10}
$$

By combining Eq. (9) and Eq. (10) we obtain the explicit form of the Ansätze. It should be noted that there is an ambiguity in the way we define the prefactors: we can either define them as objects depending on one or on two spin indices. In the former case, we end up with three independent vertices, while in the latter case, we end up with five. (The ansatz proportional to $P$ always depends only on a single spin.) The origin of this ambiguity is the freedom of pulling the sum over the internal spin in Eq. (8) into the Ansatz (leading to TPSC3) or performing the Ansatz inside the sum (leading to TPSC5). Whether both of these choices are valid in the sense that no artificial $SU(2)$ symmetry breaking is introduced has to be checked by deriving the interaction vertex induced by them. Again we stress that there is technically a much larger group of possible Ansätze due to the freedom of shifting contributions to different interaction terms.

The single spin dependent TPSC3 Ansätze are defined as (fixing the external spin to $s_1 = \uparrow$)

$$
\tilde{D}_{o_4,o_1} = D_{o_4,o_1}\frac{\langle n^\uparrow_{o_4}n^\uparrow_{o_1}\rangle + \langle n^\uparrow_{o_1}n^\downarrow_{o_4}\rangle}{\sum_{s_4}\langle n^{s_4}_{o_4}\rangle\langle n^\uparrow_{o_1}\rangle - \langle n^{\uparrow,s_4}_{o_1,o_4}\rangle\langle n^{s_4,\uparrow}_{o_4,o_1}\rangle}\,,
\tag{11}
$$

$$
\tilde{C}_{o_1,o_3} = C_{o_1,o_3}\frac{\langle n^\uparrow_{o_3}n^\uparrow_{o_1}\rangle + \langle n^{\uparrow\downarrow}_{o_3}n^{\downarrow\uparrow}_{o_1}\rangle}{\sum_{s_4}\langle n^{\uparrow,s_4}_{o_3}\rangle\langle n^{s_4,\uparrow}_{o_1}\rangle - \langle n^{s_4}_{o_1,o_3}\rangle\langle n^\uparrow_{o_3,o_1}\rangle}\,,
\tag{12}
$$

$$
\tilde{P}_{o_1,o_2} = P_{o_1,o_2}\frac{\langle n^{\uparrow\downarrow}_{o_2,o_1}n^{\downarrow\uparrow}_{o_2,o_1}\rangle}{\langle n^{\uparrow,\downarrow}_{o_2,o_1}\rangle\langle n^{\downarrow,\uparrow}_{o_2,o_1}\rangle - \langle n^\downarrow_{o_2,o_1}\rangle\langle n^\uparrow_{o_2,o_1}\rangle}\,,
\tag{13}
$$

where we dropped the dependence on the single spin as the $SU(2)$ symmetry of all quantities on the right hand side implies $X^\uparrow = X^\downarrow$ (where $X$ stands for an arbitrary ansatz). In the TPSC5

case the two-spin dependent Ansätze read

$$\tilde{D}^{s_4,s_1}_{o_4,o_1} = D_{o_4,o_1} \frac{\langle n^{s_4}_{o_4} n^{s_1}_{o_1} \rangle}{\langle n^{s_4}_{o_4} \rangle \langle n^{s_1}_{o_1} \rangle - \langle n^{s_1,s_4}_{o_1,o_4} \rangle \langle n^{s_4,s_1}_{o_4,o_1} \rangle} \, , \tag{14}$$

$$\tilde{C}^{s_1,s_3}_{o_1,o_3} = C_{o_1,o_3} \frac{\langle n^{s_1,s_3}_{o_3} n^{s_3,s_1}_{o_1} \rangle}{\langle n^{s_1,s_3}_{o_3} \rangle \langle n^{s_3,s_1}_{o_1} \rangle - \langle n^{s_1}_{o_1,o_3} \rangle \langle n^{s_1}_{o_3,o_1} \rangle} \, , \tag{15}$$

$$\tilde{P}^{s_1}_{o_2,o_1} = P_{o_2,o_1} \frac{\langle n^{s_1,\bar{s}_1}_{o_2 o_1} n^{\bar{s}_1,s_1}_{o_2,o_1} \rangle}{\langle n^{s_1,\bar{s}_1}_{o_2,o_1} \rangle \langle n^{\bar{s}_1,s_1}_{o_2,o_1} \rangle - \langle n^{\bar{s}_1}_{o_2,o_1} \rangle \langle n^{s_1}_{o_2,o_1} \rangle} \, . \tag{16}$$

In these equations, we introduced a short hand notation which allows us to drop redundant arguments, i.e. $n^{s_1,s_2}_{o_1,o_2} = c^{\dagger}_{o_2,s_2}(\tau) c_{o_1,s_1}(\tau)$ and $n^{s_1}_{o_1} = n^{s_1,s_1}_{o_1,o_1} = c^{\dagger}_{o_1,s_1}(\tau) c_{o_1,s_1}(\tau)$. Note that we dropped the time and position argument as $n$ only depends, due to translational invariance, on the difference between the arguments on the right hand side, which are identical. In principle, the two approaches should yield compatible results as long as the underlying assumptions of the approach are valid. Furthermore, it should be noted that if the initial model contains no inter-orbial hopping between orbitals which are interacting, the renormalizations of $\tilde{P}$ for both TPSC3 and TPSC5 as well as $\tilde{C}^{\uparrow,\downarrow}$ in TPSC5 are singular and thus the local and static limits cannot be captured in the standard fashion. Therefore, the question arises on how to renormalize these components in such a case. The first option is to stay at the level of plain Hartree-Fock which does not renormalize these couplings leading to an overestimation of their contribution. The second option is to use the freedom of the Ansatz and employ the same rescaling as we use for the other components. In principle, as long as the trace consistency check (equivalence of left and right hand side in Eq. (10)) is fulfilled, both variants are reasonable. However the former is expected to break down rapidly at stronger coupling, while the latter maintains the Kanamori-Brückner-like scaling [80]. We will maintain the first variant for easier comparability with earlier multi-orbital TPSC approaches [86, 89].

As is usual for TPSC [80, 82, 86], the unknown local and static TPEVs are determined self-consistently. Specifically, the TPEVs are calculated from susceptibility sum-rules, for which one requires the susceptibility, which we determine using the Bethe-Salpeter equation. The interaction which is put into the Bethe-Salpeter equation is derived as the functional derivative from the Ansatz equation – thus closing the self-consistency. In order to obtain the two-particle interaction, see Eq. (A.22) in Appendix A, one reformulates the equation of motion into an explicit equation for the self-energy, as shown here for the case of TPSC5 (the equations for TPSC3 are obtained by using the symmetries between the different components of the Ansätze):

$$\begin{aligned}
\Sigma^{s_1,s_3}_{o_1,o_3}(\tau_1;\tau_3) = \delta_{\tau_1,\tau_3} \Big( &\tilde{D}^{s'_4,s_1}_{o'_4,o_1} G^{s'_4,s'_4}_{o'_4,o'_4}(\tau_1;\tau_1) \delta_{s_1,s_3} \delta_{o_1,o_3} - \tilde{D}^{s_3,s_1}_{o_3,o_1} G^{s_1,s_3}_{o_1,o_3}(\tau_1;\tau_1) \\
&+ \tilde{C}^{s_1,s'_3}_{o_1,o_3} G^{s'_3,s'_3}_{o_1,o_3}(\tau_1;\tau_1) \delta_{s_1,s_3} - \tilde{C}^{s_1,s_3}_{o_1,o'_3} G^{s_1,s_3}_{o'_3,o'_3}(\tau_1;\tau_1) \delta_{o_1,o_3} \\
&+ \tilde{P}^{s_1,\bar{s}_1}_{o_3,o_1} G^{\bar{s}_1,\bar{s}_1}_{o_3,o_1}(\tau_1;\tau_1) \delta_{s_1,s_3} - \tilde{P}^{s_1,\bar{s}_1}_{o_3,o_1} G^{s_1,\bar{s}_1}_{o_3,o_1}(\tau_1;\tau_1) \delta_{\bar{s}_1,s_3} \Big) .
\end{aligned} \tag{17}$$

It should be noted that the self-energy we arrive at is purely spin-diagonal (due to the Green's function being spin diagonal) and $\Sigma^{\uparrow,\uparrow} = \Sigma^{\downarrow,\downarrow}$ holds due to the symmetry of the Hamiltonian which is inherited by the Ansatz.

The irreducible vertex in the direct particle-hole channel is defined as the functional derivative of the self-energy w.r.t the Green's function

$$\Gamma^{s_1 s_2 s_3 s_4}_{o_1 o_2 o_3 o_4}(\tau_1, \tau_2, \tau_3, \tau_4) = \frac{\delta \Sigma^{s_1,s_4}_{o_1,o_4}(\tau_1, \tau_4)}{\delta G^{s_3,s_2}_{o_3;o_2}(\tau_3, \tau_2)} \, . \tag{18}$$

Therefore, the vertex contains functional derivatives of the Ansätze themselves which are a priori unknown. The way to circumvent their appearance is to calculate the vertex in the

Pauli matrix basis [80], transforming spin into $n = S^0$, $S^x = S^1$, $S^y = S^2$ and $S^z = S^3$. The transformation between the Pauli ($\Gamma^{S^{i'}S^{j'}}$) and diagrammatic ($\Gamma^{s_1,s_2,s_3,s_4}$) spin space is defined by

$$\Gamma^{S^i S^j}_{o_1 o_2 o_3 o_4}(\tau_1,\tau_2,\tau_3,\tau_4) = \frac{1}{2} \sum_{s_1,s_2,s_3,s_4} \sigma^i_{s_1,s_4} \Gamma^{s_1,s_2,s_3,s_4}_{o_1 o_2 o_3 o_4}(\tau_1,\tau_2,\tau_3,\tau_4) \sigma^j_{\bar{s}_2,\bar{s}_3}. \tag{19}$$

Thus the irreducible vertex in physical spin space is

$$\Gamma^{S^i S^j}_{o_1 o_2 o_3 o_4}(\tau_1,\tau_2,\tau_3,\tau_4) = \frac{1}{2} \sum_{s_1,s_2,s_3,s_4} \sigma^i_{s_1,s_4} \frac{\delta \Sigma^{s_1,s_4}_{o_1,o_4}(\tau_1,\tau_4)}{\delta G^{s_3,s_2}_{o_3;o_2}(\tau_3,\tau_2)} \sigma^j_{s_2,s_3}. \tag{20}$$

A detailed derivation of the interaction from the self-energy is given in App. C, here we only show the final results of this calculation. For TPSC5 we find that the interaction is determined by

$$\begin{aligned}
\Gamma^{S^3 S^3}_{o_1 o_2 o_3 o_4}(\tau_1,\tau_1,\tau_1,\tau_1) &= \delta_{o_1,o_4}\delta_{o_2,o_3}\left(\tilde{D}^{\uparrow,\downarrow}_{o_3,o_1} - \tilde{D}^{\uparrow,\uparrow}_{o_3,o_1}\right) + \delta_{o_1,o_3}\delta_{o_2,o_4}\tilde{D}^{\uparrow,\uparrow}_{o_4,o_1} \\
&\quad + \delta_{o_1,o_3}\delta_{o_2,o_4}(\tilde{C}^{\downarrow,\uparrow}_{o_1,o_4} - \tilde{C}^{\uparrow,\uparrow}_{o_1,o_4}) + \delta_{o_1,o_4}\delta_{o_2,o_3}\tilde{C}^{\uparrow,\uparrow}_{o_1,o_3} \\
&\quad + \delta_{o_1,o_2}\delta_{o_3,o_4}P^{\uparrow,\downarrow}_{o_3,o_1}.
\end{aligned} \tag{21}$$

In the case of TPSC3 this equation is further simplified to

$$\Gamma^{S^3 S^3}_{o_1 o_2 o_3 o_4}(\tau_1,\tau_1,\tau_1,\tau_1) = \tilde{P}_{o_3,o_1}\delta_{o_1,o_2}\delta_{o_3,o_4} + \tilde{C}_{o_1,o_3}\delta_{o_2,o_3}\delta_{o_1,o_4} + \tilde{D}_{o_4,o_1}\delta_{o_4,o_2}\delta_{o_1,o_3}, \tag{22}$$

which is again the standard form of the inter-orbital bilinear vertex. Notably, for TPSC5 we observe that $\tilde{D}^{\uparrow\uparrow}$ and $\tilde{C}^{\uparrow\uparrow}$ do not appear independently in any of the equations, meaning that we technically do not have five but only four independent vertex components. We observe that both Ansätze fall into the required form of an $SU(2)$ symmetric interaction, justifying a posteriori that we kept both possibilities. While for the other diagonal spin components, the identical result is found, in the charge channel, no cancellation of the functional derivatives of the Ansätze exists. Hence this vertex cannot be calculated directly. Therefore, we have to resort to a two-step procedure [80]. The procedure is as follows: first one obtains the TPEV from the self-consistent solution of the spin channel, while in a next step one fits the charge channel vertex such that the sum rules derived below are fulfilled.

For the next step we need the susceptibilities. In an $SU(2)$ symmetric system, the Bethe-Salpeter equations decouple, enabling a separate evaluation of charge and spin susceptibilities, which substantially reduces the numerical effort [87]. We obtain the spin susceptibility as

$$\chi^{S^3 S^3}_{o_1,o_3,o_2,o_4}(q) = \chi^{0;S^3 S^3}_{o_1,o_3,o_{2'},o_{4'}}(q)\left(\frac{1}{1 - \Gamma^{S^3 S^3}\chi^{0;S^3 S^3}}\right)_{o_{2'}o_{4'},o_2 o_4}(q), \tag{23}$$

where we shuffled index 2, 3 and 4 compared to the Appendix C to obtain an equation in the form of a matrix-matrix product. The susceptibilities are then used to calculate the required TPEVs, which read

$$\langle n^{\uparrow}_{o_4} n^{\uparrow}_{o_1}\rangle, \qquad \langle n^{\downarrow}_{o_4} n^{\uparrow}_{o_1}\rangle, \qquad \langle n^{\uparrow\downarrow}_{o_4} n^{\downarrow\uparrow}_{o_1}\rangle = -\langle n^{\uparrow}_{o_4,o_1} n^{\downarrow}_{o_1,o_4}\rangle, \qquad \langle n^{\uparrow\downarrow}_{o_2,o_1} n^{\downarrow\uparrow}_{o_2,o_1}\rangle. \tag{24}$$

These are calculated by utilizing the orbitally-resolved sum rules of the susceptibility given by [95]

$$\sum_q \chi^{\alpha,\beta}_{o_1 o_3 o_2 o_4}(q) = \chi^{\alpha,\beta}_{o_1 o_3 o_2 o_4} = \langle T_\tau \alpha_{13}\beta_{24}\rangle - \langle \alpha_{13}\rangle\langle\beta_{24}\rangle. \tag{25}$$

Here, $\alpha$ and $\beta$ are either spin operators or density operators. For spin operators in an $SU(2)$ symmetric system $\langle\alpha_{13}\rangle$ has to be zero, so that we can drop the single particle contribution in the following.

First, we rewrite the sum rule for the spin-$z$ susceptibility

$$\chi^{S^3S^3}_{o_1,o_3,o_2,o_4} = \sum_\sigma \langle n^\sigma_{o_1o_3} n^\sigma_{o_2o_4} - n^\sigma_{o_1o_3} n^{\bar\sigma}_{o_2o_4}\rangle \tag{26}$$

$$= 2\langle n^\uparrow_{o_1o_3} n^\uparrow_{o_2o_4} - n^\uparrow_{o_1o_3} n^\downarrow_{o_2o_4}\rangle . \tag{27}$$

By differentiating cases with pairwise identical indices we arrive at the following set of equations

$$1=3=2=4 \quad\Longrightarrow\quad \chi^{S^3S^3}_{o_1,o_1,o_1,o_1} = 2\langle n^\uparrow_{o_1}\rangle - 2\langle n^\uparrow_{o_1} n^\downarrow_{o_1}\rangle , \tag{28}$$

$$1=3\neq 2=4 \quad\Longrightarrow\quad \chi^{S^3S^3}_{o_1,o_1,o_2,o_2} = 2\langle n^\uparrow_{o_1} n^\uparrow_{o_2} - n^\uparrow_{o_1} n^\downarrow_{o_2}\rangle , \tag{29}$$

$$1=4\neq 3=2 \quad\Longrightarrow\quad \chi^{S^3S^3}_{o_1,o_2,o_2,o_1} = 2\langle n^\uparrow_{o_2}\left(1-n^\uparrow_{o_1}\right)\rangle - 2\langle n^\uparrow_{o_1o_2} n^\downarrow_{o_2o_1}\rangle , \tag{30}$$

$$1=2\neq 3=4 \quad\Longrightarrow\quad \chi^{S^3S^3}_{o_1,o_2,o_1,o_2} = 2\langle n^{\uparrow\downarrow}_{o_1o_2} n^{\downarrow\uparrow}_{o_1o_2}\rangle , \tag{31}$$

where we utilized the Pauli principle in the special case where all indices are identical. However, these equations are not sufficient to fix all unknowns. To do so, we use the fact that the susceptibility is $SU(2)$ symmetric, i.e. $\chi^{S^1S^1} = \chi^{S^2S^2} = \chi^{S^3S^3}$. This allows us to obtain more sum rules by evaluating the expression for the other components (note that $S^{1/2}$ means $S^1$ for the upper sign and $S^2$ for the lower sign):

$$\chi^{S^{1/2}S^{1/2}}_{o_1,o_3,o_2,o_4} = \pm\langle\left(c^\dagger_{o_3\downarrow}c_{o_1\uparrow} \pm c^\dagger_{o_3\uparrow}c_{o_1\downarrow}\right)\left(c^\dagger_{o_4\downarrow}c_{o_2\uparrow} \pm c^\dagger_{o_4\uparrow}c_{o_2\downarrow}\right)\rangle \tag{32}$$

$$= \pm\left(\langle c^\dagger_{o_3\downarrow}c_{o_1\uparrow}c^\dagger_{o_4\downarrow}c_{o_2\uparrow}\rangle + \langle c^\dagger_{o_3\uparrow}c_{o_1\downarrow}c^\dagger_{o_4\uparrow}c_{o_2\downarrow}\rangle\right. \tag{33}$$

$$\left. \pm\langle c^\dagger_{o_3\downarrow}c_{o_1\uparrow}c^\dagger_{o_4\uparrow}c_{o_2\downarrow}\rangle \pm \langle c^\dagger_{o_3\uparrow}c_{o_1\downarrow}c^\dagger_{o_4\downarrow}c_{o_2\uparrow}\rangle\right) ,$$

which, analogously to the spin-$z$ susceptibility, are inspected for pair-wise identical indices:

$$1=3\neq 2=4 \quad\Longrightarrow\quad \chi^{S^{1/2}S^{1/2}}_{o_1,o_1,o_2,o_2} = 2\left(\pm\langle n^{\downarrow\uparrow}_{o_1} n^{\downarrow\uparrow}_{o_2}\rangle - \langle n^\downarrow_{o_2o_1} n^\uparrow_{o_1o_2}\rangle\right) , \tag{34}$$

$$1=4\neq 3=2 \quad\Longrightarrow\quad \chi^{S^{1/2}S^{1/2}}_{o_1,o_2,o_2,o_1} = \langle n_{o_2}\rangle + 2\left(\mp\langle n^{\downarrow\uparrow}_{o_1} n^{\downarrow\uparrow}_{o_2}\rangle - \langle n^\downarrow_{o_2} n^\uparrow_{o_1}\rangle\right) , \tag{35}$$

$$1=2\neq 3=4 \quad\Longrightarrow\quad \chi^{S^{1/2}S^{1/2}}_{o_1,o_2,o_1,o_2} = 2\langle n^{\uparrow\downarrow}_{o_1,o_2} n^{\downarrow\uparrow}_{o_1,o_2}\rangle . \tag{36}$$

Due to $SU(2)$ symmetry, all expectation values with a $\pm$ in front have to be zero. Thereby, we have a system with four unknowns and six equations, which at first glance seems to be overdetermined. However, one realizes that the limit $1=2\neq 3=4$ gives the same sum rule irrespective of the spin-channel, thus leaving us with four equations for three unknowns:

$$1=3\neq 2=4 \quad\Longrightarrow\quad \chi^{S^3S^3}_{o_1,o_1,o_2,o_2} = 2\langle n^\uparrow_{o_1} n^\uparrow_{o_2} - n^\uparrow_{o_1} n^\downarrow_{o_2}\rangle = -2\langle n^\downarrow_{o_2o_1} n^\uparrow_{o_1o_2}\rangle , \tag{37}$$

$$1=4\neq 3=2 \quad\Longrightarrow\quad \chi^{S^3S^3}_{o_1,o_2,o_2,o_1} = 2\langle n^\uparrow_{o_2}(1-n^\uparrow_{o_1})\rangle - 2\langle n^\uparrow_{o_1o_2} n^\downarrow_{o_2o_1}\rangle \tag{38}$$

$$= \langle n_{o_2}\rangle - 2\langle n^\downarrow_{o_2} n^\uparrow_{o_1}\rangle .$$

These equations are not independent – as long as we fulfill three of the equations, the fourth one is automatically fulfilled. Hence, the system is not over-determined and the TPEVs can be

calculated as

$$\langle n_{o_2}^{\downarrow} n_{o_1}^{\uparrow} \rangle = \frac{\langle n_{o_2} \rangle - \chi_{o_1,o_2,o_2,o_1}^{S^3 S^3}}{2}, \tag{39}$$

$$\langle n_{o_2}^{\uparrow} n_{o_1}^{\uparrow} \rangle = \langle n_{o_2}^{\downarrow} n_{o_1}^{\uparrow} \rangle + \frac{\chi_{o_1,o_1,o_2,o_2}^{S^3 S^3}}{2}, \tag{40}$$

$$\langle n_{o_1 o_2}^{\uparrow} n_{o_2 o_1}^{\downarrow} \rangle = -\frac{\chi_{o_1,o_1,o_2,o_2}^{S^3 S^3}}{2}, \tag{41}$$

$$\langle n_{o_1,o_2}^{\uparrow\downarrow} n_{o_1,o_2}^{\downarrow\uparrow} \rangle = \frac{\chi_{o_1,o_2,o_1,o_2}^{S^3 S^3}}{2}. \tag{42}$$

In summary, the self-consistency loop consists of the Ansatz equations for TPSC3 (Eq. (11,12,13)) or TPSC5 (Eq. (14,15,16)), from which the vertex follows in the form of Eq. (22) or Eq. (21). The vertex in turn determines the susceptibility, calculated according to Eq. (23), from which the TPEVs can be extracted utilizing the sum rules (Eq. (39,40,41,42)). This loop can be solved in various ways – in its core it is a minimization or multidimensional root finding problem. Once a minimum is found, the TPEVs are used to determine the charge vertex by fitting to the sum rules in the charge sector. These are derived analogously to the spin sum rules and read for pair-wise identical indices:

$$1 = 3 \neq 2 = 4 \quad \implies \quad \chi_{n_{o_1 o_1} n_{o_2 o_2}} = 2 \langle n_{o_1}^{\uparrow} n_{o_2}^{\uparrow} + n_{o_1}^{\uparrow} n_{o_2}^{\downarrow} \rangle - \langle n_{o_1} \rangle \langle n_{o_2} \rangle, \tag{43}$$

$$1 = 4 \neq 3 = 2 \quad \implies \quad \chi_{n_{o_1 o_2} n_{o_2 o_1}} = 2 \langle n_{o_2}^{\uparrow} (1 - n_{o_1}^{\uparrow}) \rangle + 2 \langle n_{o_1 o_2}^{\uparrow} n_{o_2 o_1}^{\downarrow} \rangle \tag{44}$$
$$- \langle n_{o_1 o_2} \rangle \langle n_{o_2 o_1} \rangle,$$

$$1 = 2 \neq 3 = 4 \quad \implies \quad \chi_{n_{o_1 o_2} n_{o_1 o_2}} = -2 \langle n_{o_1 o_2}^{\uparrow\downarrow} n_{o_1 o_2}^{\downarrow\uparrow} \rangle - \langle n_{o_1 o_2} \rangle \langle n_{o_1 o_2} \rangle. \tag{45}$$

Since we have only three equations irrespective of the type of Ansatz we picked, this part of the calculation is identical for both TPSC3 and TPSC5, as long as we fit a charge vertex parameterized in the inter-orbital bilinear form.

In the following we analyze this approach both numerically and analytically and discuss its strengths and shortcomings.

## 2.1 Differences to earlier approaches

Multi-orbital TPSC formalisms were already presented in Ref. [89] and Ref. [90]. Here, we briefly discuss the differences between the proposed approach and these earlier attempts of generalizing TPSC to a multi-orbital setting. First and foremost, the proposed procedure here is fully self-consistent, i.e. no additional symmetry constraints have to be enforced beyond the structure of the bare vertex. The self-consistent double occupancies seem to cure the issue of negative components of the charge vertex - thus the internal consistency check is fulfilled as long as both minimizations converge to a solution. However, utilizing more sum rules complicates the numerical root search, so that the computational cost of the present approach is higher. We include both the particle-hole symmetrized (see App. D) and the usual version of the Ansätze in our implementation. However, in this paper, we only consider half-filled systems, so that the results between the particle-hole symmetry enforcing and the usual Ansatz do not differ. Apart from these differences, the sum rules utilized in Ref. [89] and Ref. [90] are a subset of the sum-rules employed here, see App. B.1, and we checked that by constraining the equations we can reproduce the previous results.

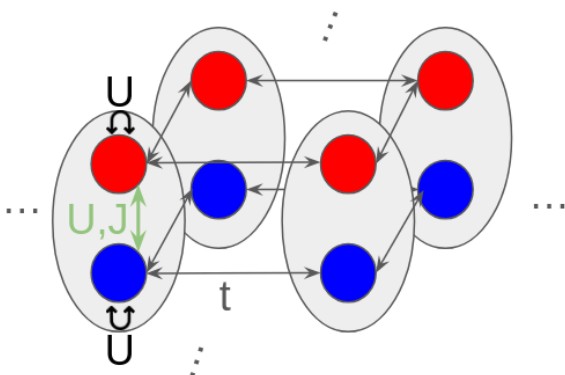

Figure 1: Pictorial view of the prototypical two-orbital Hubbard model considered in this work.

## 3 Benchmarking

Before benchmarking the variants numerically, we will try to provide an analytical understanding of when the approach is suitable and when it is not. The models utilized in these benchmarks are variations of a multi-orbital Hubbard model, where each site contains two orbitals, see Fig. 1. The Hamiltonian of this model reads

$$\hat{H} = \sum_{s,\langle i,j \rangle,o} t c^{\dagger}_{o,s,i} c_{o,s,j} + \hat{H}_{HK} \,, \tag{46}$$

where $i, j$ are lattice sites and $o$ labels the orbitals at each site. Here we introduced a Hubbard-Kanamori interaction, which has the following form

$$H^{HK} = U \hat{n}^{\uparrow}_{o_{1'}} \hat{n}^{\downarrow}_{o_{1'}} + \sum_{o_1 \neq o_2} \left( (U - 2J) \hat{n}^{\uparrow}_{o_1} \hat{n}^{\downarrow}_{o_2} + (U - 3J) \hat{n}^{s'}_{o_1} \hat{n}^{s'}_{o_2} \right.$$
$$\left. - J \hat{c}^{\dagger}_{o_1,\uparrow} \hat{c}_{o_1,\downarrow} \hat{c}^{\dagger}_{o_{2'},\downarrow} \hat{c}_{o_2,\uparrow} + J \hat{c}^{\dagger}_{o_1,\uparrow} \hat{c}^{\dagger}_{o_1,\downarrow} \hat{c}_{o_2,\downarrow} \hat{c}_{o_2,\uparrow} \right) \,. \tag{47}$$

We again stress that primed variables are implicitly summed over, here we made one of these sum explicit to catch the special case of equal orbitals. To extract the form of the vertex components $P$, $C$ and $D$, one first determines $U_{o_1,o_2,o_3,o_4}$ by fixing a specific spin configuration, e.g. $\uparrow\downarrow\uparrow\downarrow$, in the full interaction tensor $V^{s_1,s_2,s_3,s_4}_{o_1,o_2,o_3,o_4}$. In a second step we utilize the Kronecker deltas in Eq. (4) to decompose $U$ into the components. The on-site interaction is put into $D$. The three components read

$$D_{o_1,o_4} = U - (1 - \delta_{o_1,o_4})2J \,, \tag{48}$$

$$C_{o_1,o_3} = (1 - \delta_{o_1,o_3})J \,, \tag{49}$$

$$P_{o_1,o_3} = (1 - \delta_{o_1,o_3})J \,. \tag{50}$$

If not mentioned otherwise, the two orbitals are not kinetically coupled, we set $t = 1$ and give all other quantities relative to $t$. The orbitals have a coupling on the two-particle level due to the Hubbard-Kanamori interaction, which is parameterized by the on-site interaction $U$ and the Hunds coupling $J$. Such a model and its variations have been studied extensively in former works [19, 90, 95], to which we will compare our results. To analyze this model analytically, we consider two limiting cases, the one in which no Hunds coupling is present and the case in which the Hunds-coupling is $U/3$.

## 3.1 Model with pure density-density interactions

First, let us consider the multi-orbital model described above with density-density interactions only. Furthermore, we assume half-filling, i.e. $\langle n_{o_1,s}\rangle = 0.5$, such that we have one particle per orbital. Due to the absence of inter-orbital hoppings the non-interacting susceptibility has a block diagonal form:

$$\chi^0_{o_1,o_2,o_3,o_4}(\boldsymbol{q},\omega) = \sum_{\boldsymbol{k},\nu} G_{o_1 o_4}(\boldsymbol{k},\nu) G_{o_2 o_3}(\boldsymbol{k}-\boldsymbol{q},\nu-\omega) = \delta_{o_1,o_4}\delta_{o_2,o_3}\chi^0(\boldsymbol{q},\omega),\qquad(51)$$

where $\chi^0(\boldsymbol{q},\omega)$ is the susceptibility of the respective single-band model, $\boldsymbol{q}$ is the momentum and $\omega$ the Matsubara frequency. Furthermore, we consider a pure density-density type interaction with identical intra- ($U$) and inter-orbital ($U'$) Hubbard interaction (zero Hund's coupling $J$):

$$H_{\text{int}} = \sum_{i,o_1,o_2,s} U n_{i,o_1,s} n_{i,o_2,\bar{s}} + \left(1-\delta_{o_1 o_2}\right) U' n_{i,o_1,s} n_{i,o_2,s}.\qquad(52)$$

This interaction in combination with the kinetic term leads to a vanishing $\Sigma^0$ (see App. (A.2) Eq. A.39 or Ref. [91]). Since $J = 0$ there is no spin selectivity and we thus expect $\langle n_0^\uparrow n_1^\uparrow\rangle = \langle n_0^\uparrow n_1^\downarrow\rangle$. Furthermore, since $U = U'$, there is no energy difference for double occupying the same or different orbitals, and hence $\langle n_0^\uparrow n_1^\downarrow\rangle = \langle n_0^\uparrow n_0^\downarrow\rangle$. This also implies that the double occupancy $\langle n_0^\uparrow n_0^\downarrow\rangle$ should decrease less rapidly with $U$ than in the case with $U' = 0$ (which then is equivalent to the single-orbital Hubbard model). With this in mind we now turn to TPSC.

For a pure density-density interaction, the only non-zero Ansatz (TPSC3 and TPSC5 are giving identical results in this specific case, so we discuss TPSC3 here) is the one for $\tilde{D}$:

$$\tilde{D}_{o_4,o_1} = D_{o_4,o_1} \frac{\sum_{s_4} \langle n_{o_4}^{s_4} n_{o_1}^{s_1}\rangle}{\sum_{s_4} n_{o_4}^{s_4} n_{o_1}^{s_1} - n_{o_1 o_4}^{s_1 s_4} n_{o_4 o_1}^{s_4 s_1}}.\qquad(53)$$

As we are at half-filling and have no inter-orbital coupling, the Hamiltonian is diagonal in orbital space, i.e. $\langle c_{o_1,s_1}^\dagger c_{o_2,s_2}\rangle = 0.5\,\delta_{o_1,o_2}\delta_{s_1,s_2}$. Therefore the denominator simplifies to

$$\sum_{s_4} n_{o_4}^{s_4} n_{o_1}^{s_1} - n_{o_1 o_4}^{s_1 s_4} n_{o_4 o_1}^{s_4 s_1} = 0.5 - 0.25\,\delta_{o_1,o_4}.\qquad(54)$$

The numerator is obtained via the Ansatz equations and thereby via the interacting susceptibility. Inserting the explicit form of the vertex derived from the Ansatz, as well as the non-interacting susceptibility, we arrive at

$$\chi_{o_1,o_2,o_3,o_4}(\boldsymbol{q},\omega) = \sum_{o_{1'},o_{3'}} \delta_{o_1,o_{3'}}\delta_{o_{1'},o_3}\chi^0(\boldsymbol{q},\omega)\left(\frac{1}{\hat{1}-\hat{\tilde{D}}\hat{\chi}^0(\boldsymbol{q},\omega)}\right)_{o_{1'},o_2,o_{3'},o_4},\qquad(55)$$

where the fraction denotes an inversion. The matrix we need to invert reads in index notation

$$\left(\hat{1}-\hat{\tilde{D}}\hat{\chi}^0(\boldsymbol{q},\omega)\right)_{o_1,o_2,o_3,o_4} = \delta_{o_1,o_3}\delta_{o_2,o_4} - \delta_{o_1,o_3}\delta_{o_{2'},o_{4'}} D_{o_1,o_{2'}}\chi^0(\boldsymbol{q},\omega)\delta_{o_{2'},o_4}\delta_{o_2,o_{4'}},\qquad(56)$$

which has a block diagonal structure. Utilizing that the inverse matrix of a block diagonal matrix is again block diagonal, we arrive at

$$\chi_{o_1,o_3,o_2,o_4}(\boldsymbol{q},\omega) = \delta_{o_1,o_4}\delta_{o_2,o_3}\chi^0(\boldsymbol{q},\omega)\frac{1}{1-\chi^0(\boldsymbol{q},\omega)\tilde{D}}.\qquad(57)$$

We note that the interacting susceptibility is identical to the one of a single-orbital model with the same renormalized interaction value. The only non-zero components of the susceptibility are $\chi_{1111}$ and $\chi_{1221}$. Thus, from Eq. (39) and Eq. (40) it immediately follows that $\langle n_0^\uparrow n_1^\uparrow \rangle = \langle n_0^\uparrow n_1^\downarrow \rangle = \langle n_0^\uparrow n_0^\downarrow \rangle$, as we expected from the arguments above. Inserting these findings into the Ansatz equations yields

$$\tilde{D}_{o_4,o_1} = U \frac{\langle n_{o_1}^\uparrow n_{o_4}^\uparrow \rangle \left(1 - \delta_{o_1,o_4}\right) + \langle n_{o_1}^\uparrow n_{o_4}^\downarrow \rangle}{0.5 - 0.25 \delta_{o_1,o_4}} = 4U \langle n_{o_1}^\uparrow n_{o_4}^\downarrow \rangle \; . \tag{58}$$

Due to the symmetries we found, the right hand side is orbitally independent and gives the same equation for the renormalized interactions as the single-orbital case [80,82]. This result is also independent of what filling we initially chose and is a general feature of the considered model.

Therefore, in contrast to our expectations, the specific form of the susceptibility implies that the converged TPSC loop will result in the same double occupancy as the one for the one-orbital model. In other words, adding an inter-orbital interaction of the same strength as the bare interaction does not make a difference for the double occupation predicted by the method. This is in contrast to the physically expected result discussed above – in the single-orbital model in the strong coupling limit, we would expect zero double occupancy. However in the two-orbital model, different states in which no orbital is doubly occupied and states in which one orbitals is doubly occupied are degenerate in energy, so the double occupancy should converge towards a nonzero value.

To pinpoint what TPSC is missing in this case we go back one step and consider a molecular system. Here, multi-orbital TPSC is essentially amounting to a small correction with respect to the underlying Hartree-Fock treatment, meaning we essentially benchmark the validity of the Hartree-Fock decoupling. The simplest such system we can write down are two coupled dimers, visualized for a four dimer case in Fig. 1, without Hund's coupling $J$. The hopping $t$ again is set to 1 and $U$ is varied. We compare the results between ED and TPSC, or better said Hartree-Fock. The behavior described above is reproduced, see Fig. 2a. Further, since the susceptibility does not gain any strong $\tau$ dependence, implying that the frequency dependence also does not change drastically, as shown in Fig. 2b. Therefore, we expect the vertex to be still reasonably well described by the static limit. The questions therefore are, first, where does the deviation stem from and second, why does this issue occur here but not in the single-orbital model?

Let us first answer the second question. In the single-orbital case we fix the local and static expectation values to be the exact ones as determined by the sum rules. Crucially, the single site contains no further internal structure meaning that the exact local expectation value is always proportional to the non-interacting one. Thus representing this expectation value locally by a renormalized Hartree-Fock expectation value works, i.e., we can always get the correct value of the double occupancy by multiplying the non-interacting double occupancy with a single number.

Multi-orbital systems have additional degrees of freedom, which enter the local and static contribution. However, we still approximate the TPEVs in Eq. (8) by a Hartree-Fock decoupling. To make matters simple, let us first discuss the zero temperature limit. In this limit, the perturbative many-body theory Hartree-Fock variant we use to decouple the expectation values becomes equivalent to the variational Hartree-Fock variant. Thus, let us consider the ladder one for now. Variational Hartree-Fock approximates the ground-state by a single-Slater determinant. However, in cases in which the ground state is degenerate, *all* of the degenerate states of the local Hilbert-space influence the expectation values. Thereby, the TPEV's we obtain from Hartree-Fock are biased by the introduction of an artificial preference to a single state which was selected arbitrarily. Of course, TPSC is a finite temperature method - so while

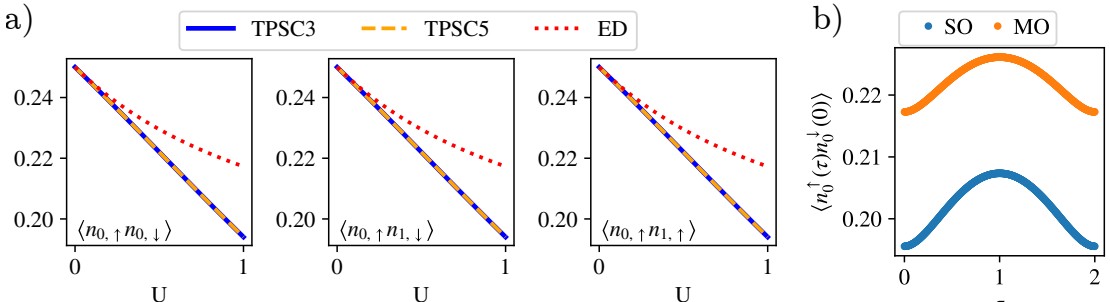

Figure 2: (a) Comparison between TPSC and ED at $\beta = 2$ for different $U$ at $J = 0$. We observe the same phenomenology as expected from our analytical analysis. (b) Time dependence of the density-density correlator for a single (SO) and a two-orbital (MO) two-site Hubbard model from ED at $\beta = 2$. No enhancement of the time dependence is visible, and hence this cannot explain the larger deviation to DMFT in the multi-orbital case.

this argument explains what we observe at sufficiently low temperatures, it can only give us a hint about the physical origin. To better understand the issue let us consider a single dimer (i.e. a single site in Fig. 1) at half-filling with a Hubbard-Kanamori interaction. For simplicity, let us call the two orbitals $a$ and $b$. The Hamiltonian then contains only interaction terms and reads

$$H = \sum_{i \in \{a,b\}} U \hat{n}_{o_i}^{\uparrow} \hat{n}_{o_i}^{\downarrow} + \left( (U - 2J)\hat{n}_a^{\uparrow} \hat{n}_b^{\downarrow} + (U - 3J)\hat{n}_a^{s'} \hat{n}_b^{s'} - J \hat{c}_{a,\uparrow}^{\dagger} \hat{c}_{a,\downarrow} \hat{c}_{b,\downarrow}^{\dagger} \hat{c}_{b,\uparrow} + J \hat{c}_{a,\uparrow}^{\dagger} \hat{c}_{a,\downarrow}^{\dagger} \hat{c}_{b,\downarrow} \hat{c}_{b,\uparrow} \right). \quad (59)$$

This Hamiltonian can be readily diagonalized in the subspace containing two electrons ($|\uparrow_a, \downarrow_a\rangle$, $|\uparrow_a, \downarrow_b\rangle$, $|\uparrow_b, \downarrow_a\rangle$, $|\uparrow_a, \uparrow_b\rangle$, $|\downarrow_a, \downarrow_b\rangle$, $|\uparrow_b, \downarrow_b\rangle$). We find three different groups of eigenstates: The first one is three-fold degenerate with eigenvalue $U - 3J$ and spanned by $|\uparrow_a, \uparrow_b\rangle$, $|\downarrow_a, \downarrow_b\rangle$ and $(|\uparrow_a, \downarrow_b\rangle - |\uparrow_b, \downarrow_a\rangle)/\sqrt{2}$. These three states form the $S = 1$ sector. Next, we do have two states with eigenvalue $U - J$, $(|\uparrow_a, \downarrow_b\rangle + |\uparrow_b, \downarrow_a\rangle)/\sqrt{2}$ and $(|\uparrow_a, \downarrow_a\rangle - |\uparrow_b, \downarrow_b\rangle)/\sqrt{2}$. And lastly, a single eigenstate with eigenvalue $U + J$, $(|\uparrow_a, \downarrow_a\rangle + |\uparrow_b, \downarrow_b\rangle)/\sqrt{2}$.

In this specific case, in the absence of $J$ we directly observe that the local self-energy, see Eq. 17, becomes a diagonal matrix independent of the orbital, as the Fock term vanishes (this is also true for the case discussed in Sec. 3). Thus, fixing the particle number absorbs the Hartree-Fock correction into the chemical potential. Therefore, the Hartree-Fock solution at fixed particle number is the same as the non-interacting model - directly explaining why above we observed that adding the inter-orbital interactions was reproducing the single-orbital case. Since the TPSC ansatz in this limit starts from a point indistinguishable from the single-orbital case due to the approximations made, it generates the same correction and follows the same trajectory.

## 3.2 Analysis of the model with $J = U/3$

If $J = U/3$, the same-spin density-density interaction becomes exactly zero. This choice is interesting, as it is the limiting case where the approach in Ref. [90] and TPSC5 show the best agreement compared to DMFT, see Fig. 3. What is the origin of this apparent improvement?

First, we should note that this is not a magic value at which the results suddenly agree, but there appears to be a steady improvement with increasing $J$. Second, we observe that for $J = U/3$ the same-spin interaction is exactly 0 and stays at that value throughout the calculation in TPSC5. In TPSC3 this interaction will gain a nonzero value, leading to the instability we observe in Fig. 3.

Since we found in the last section that the problems arising can be traced back to the Hartree-Fock ansatz, the question is whether we can again understand the improvement as an inherited effect. To test this, we again consider the half-filled dimer described above. At $T = 0$ we again have to consider degeneracies: We found three groups of eigenvalues above, whose eigenvalues split with increasing $J$. Thus, while the point $U = 3J$ is not special, the separation between the excited states and the ground state sector grows linearly with $J$. Furthermore, the $S = 1$ subspace is special, as two states only contribute to a single TPEV ($\langle n_0^\uparrow n_1^\uparrow \rangle$) and the other state essentially behaves like the single-orbital model. Thus, at $T = 0$ we would expect already better performance of the ansatz as the degeneracy is lifted. Therefore, one should be able to approximate the exact TPEV's by a scaling in all components but the $\langle n_0^\uparrow n_1^\uparrow \rangle$ one, which is incorrectly estimated because of the degenerate states being incorrectly represented in the perturbation theory. This is exactly what is observed for both TPSC5 and in Ref. [90]. How is this lifting of the ground state degeneracy influencing the behavior at nonzero temperature? - At the first glance, it does not directly, as still the introduction of the Hund's coupling merely changes the numerical value of the self-energy but it still remains a diagonal matrix without any orbital selectivity. Therefore, the starting point is again the non-interacting model, just as before. However, introducing $J$ allows for a differentiation of equal orbital and unequal orbital cases leading to a better correction within TPSC. Thus we conclude that in the end the improvement is related to the structure of the equation changing by the introduction of a Hund's coupling. Further, we note that we can analogously understand the improvement by considering the number of degenerate ground-states within the local Hilbert-space.[1]

From these observations and the connection at $T = 0$ to variational Hartree-Fock, we can extract guiding principles of when to expect TPSC to work reliably in a multi-orbital setting and when not: If there is a gap in the local spectrum between the ground state and the excited states, TPSC should perform better than if the states are closer in energy. Furthermore, when analyzing the nature of the states, we can extract which TPEV is expected to deviate strongly and which not. In other words, in cases in which Hartree-Fock produces a sensible starting point, TPSC is expected to perform well. Whether this is the case or not strongly depends on the validity of approximating the self-energy to be static. On the other hand, at very low temperatures, we expect the envelope diagram to give a sizable contribution near the Fermi-level inducing a non-local and non-static irreducible vertex [107, 108]. Additionally, it was shown in an RG analysis that in the low temperature limit, the vertex is only RG relevant at the Fermi-surface again implying a momentum and Frequency dependence strongly confined in both spaces [28]. Therefore, the local and static approximation of the vertex becomes more and more inappropriate at lower temperatures which is why TPSC is expected to fail at very low temperatures.

In summary, the regimes in which multi-orbital TPSC is guaranteed to work well are the region in which Hartree-Fock in itself provides a sensible starting point, e.g. the weak-coupling limit and cases in which the self-energy is well approximated by a static function. Furthermore, the vertex has to be approximately local and static. The former criterion we can try to gauge by considering the spectrum of the local Hilbert-space due to the connection to the variational Hartree-Fock flavor. The latter one is not clear a priori and requires other numerical simulations to be fully gauged. Notably, the issue described above is partially resolved when starting from DMFT TPEVs [91, 95], since fixing the double occupancies through an external source prevents the negative feedback from a wrong starting point.

---

[1]That this analogy works so well is a consequence of the true many-body Green's function being better representable by Hartree-Fock in the limit of a gapped ground state.



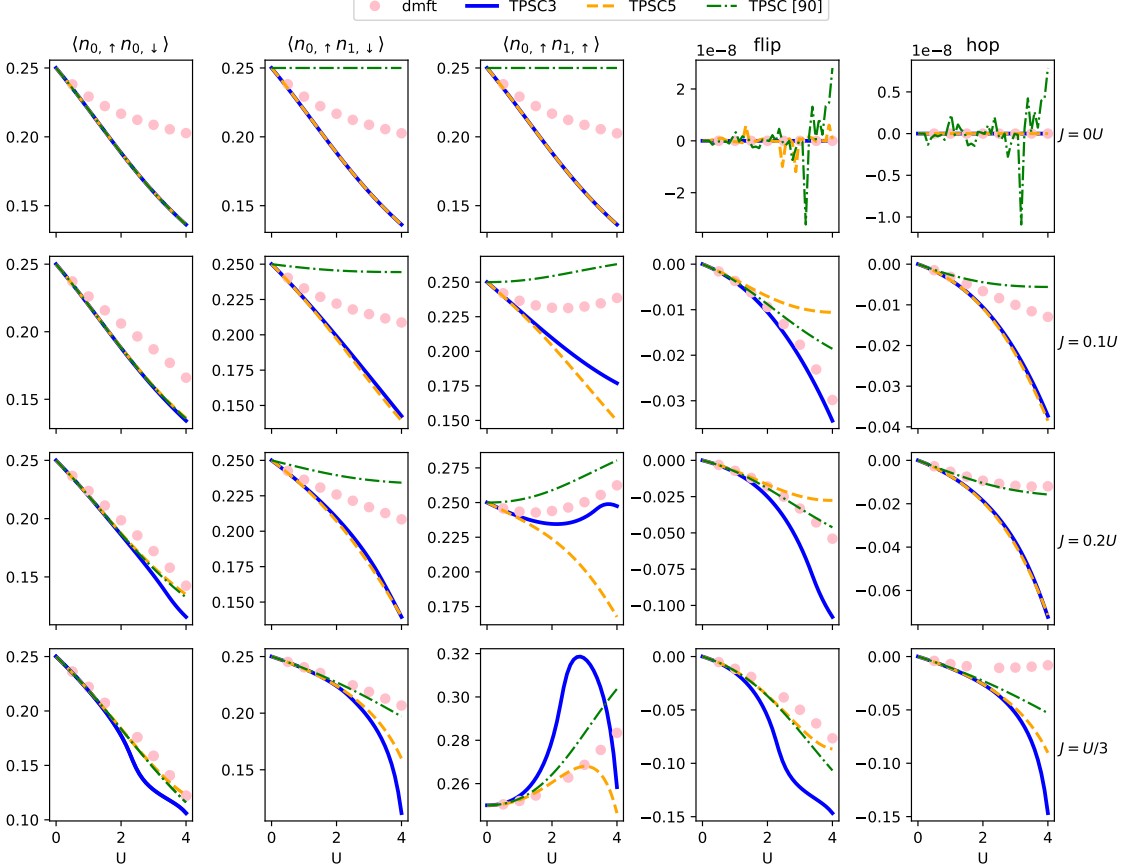

Figure 3: Comparison between density-density correlations obtained with TPSC3 (green), TPSC5 (red), TPSC from Ref. [90] (blue) and DMFT (pink dots) for the two-orbital Hubbard model defined in Eq. (46) at $\beta = 2$, for different $U$ and $J$ values.

## 3.3 Numerical benchmarks

Keeping the above described shortcomings in mind, we are now exploring the range of applicability of multi-orbital TPSC by testing the reliability of TPSC3 and TPSC5 through comparison to more accurate numerical approaches. In a first step, we exclusively aim at validating the quality of the predicted TPEVs, as these form the basis for the TPSC approach. For this we compare to both DMFT and D-Trilex. It should be stressed that while neither of the approaches are exact, both methods have proven to be suitable for the considered parameter regime [19,50].

### 3.3.1 Comparison to DMFT - TPEV's

First, we compare the results from TPSC to DMFT. For this, we consider the same model as Ref. [90], which we already introduced above, see Eq. (46). The inverse temperature is set to $\beta = 2$, thus no sharp features in momentum space appear. We run the simulations on a $24 \times 24$ momentum mesh and utilize the sparse-ir [109–111] to compress the single particle Green's function with $\omega_{\max} = 30$ and a tolerance of $10^{-12}$ for the singular values. For completeness, we also include the results from the implementation proposed in Ref. [90] (here named TPSC). The DMFT calculations are performed using w2dynamics [64].

The considered model can be seen as a worst-case scenario as in the absence of inter-orbital hopping the off-diagonal occupations are guaranteed to be zero. Hence, the $P$-channel as well as the opposite spin $C$-channel (in the case of TPSC5) Ansatz equations are ill-defined, making

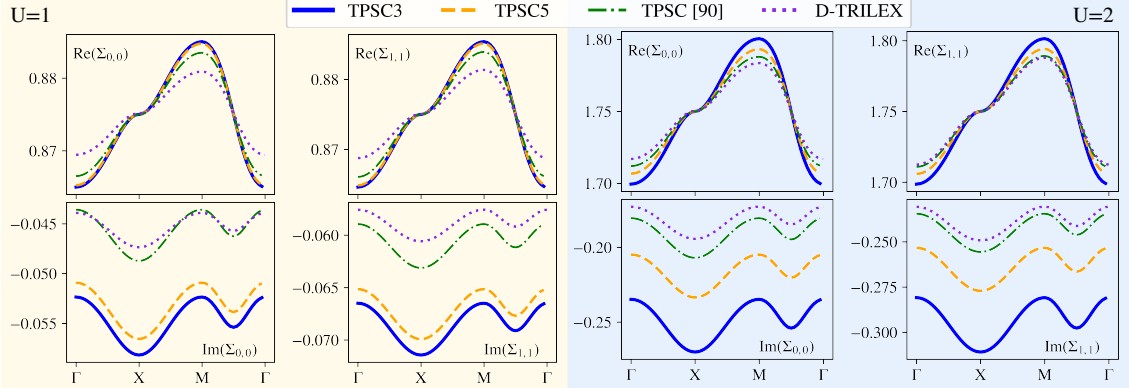

Figure 4: Comparison of different self-energy components at the first Matsubara frequency between TPSC3 (blue), TPSC5 (orange), TPSC from Ref. [90] (green) and D-TRILEX (purple) at different $U$ indicated by different background colors and $J = 0.25U$. The inverse temperature is fixed to $\beta = 2$.

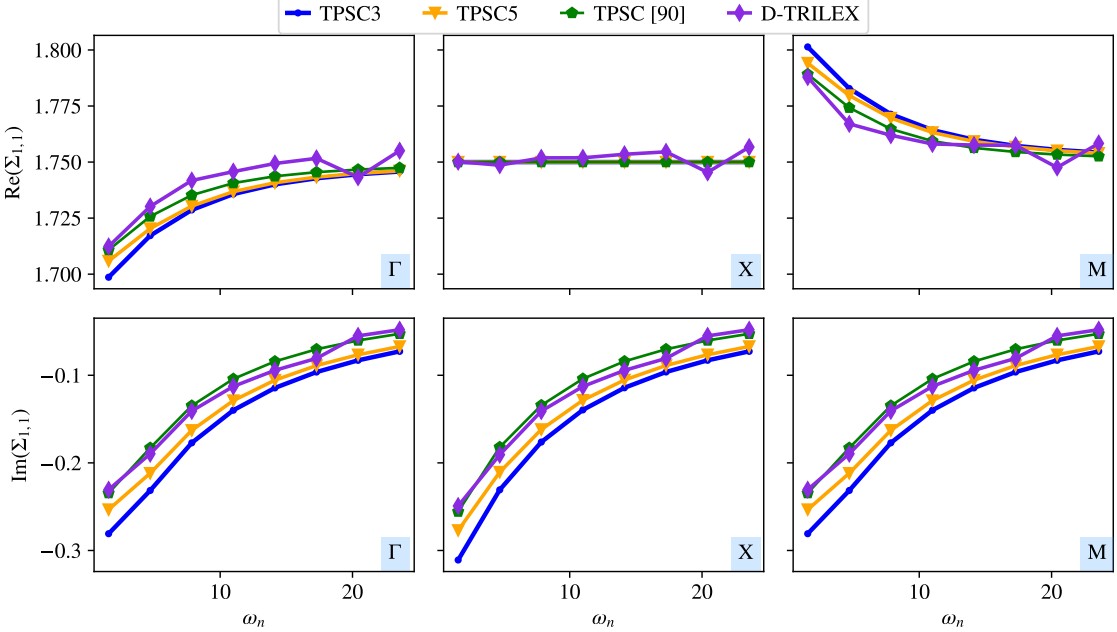

Figure 5: Comparison of $\Sigma_{1,1}$ at different momentum points (indicated in the bottom right corner) as a function of Matsubara frequency between TPSC3 (blue), TPSC5 (orange), TPSC from Ref. [90] (green) and D-TRILEX (purple) at $U = 2$ and $J = 0.25U$. The inverse temperature is fixed to $\beta = 2$.

a renormalization of the corresponding vertices impossible. In other words, the corresponding Hartree-Fock decouplings always results in a net-zero contribution of these terms to the self-energy. Interestingly, even though the Ansatz equations might break down, the derivation remains valid – we can still renormalize these components, albeit in a less controlled fashion. We found that fixing the renormalization of the channels to that of the one channel in which the Ansatz is not ill-defined works for a wide range of parameters.

Both proposed TPSC approaches show promising results in the weak-coupling regime, but the deviation from DMFT rapidly increases with stronger interactions. The error is largest in the absence of Hund's coupling and becomes smaller for larger $J$, as expected from our

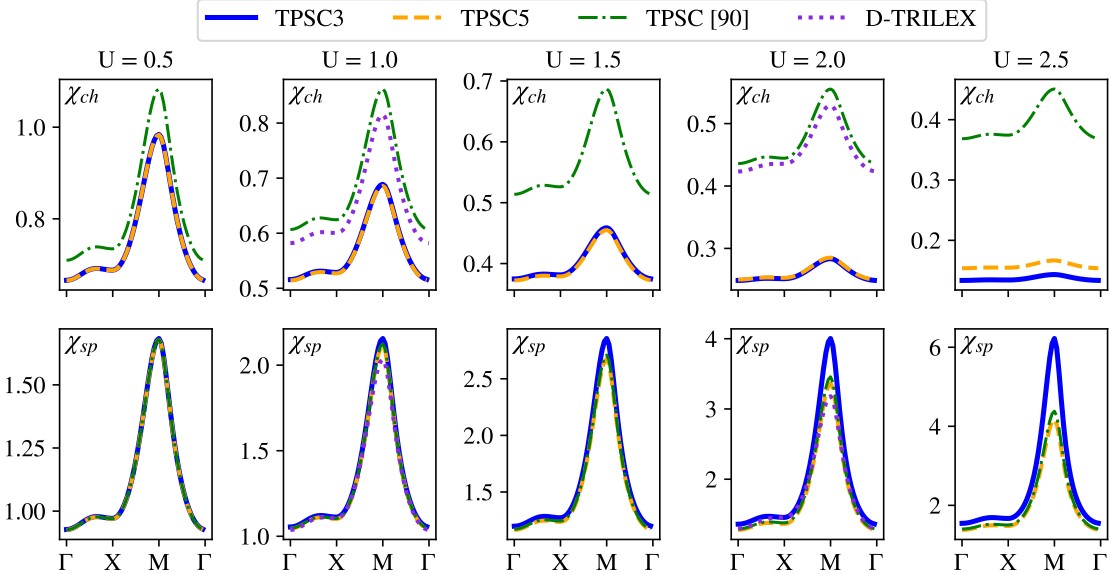

Figure 6: Comparison of the charge (upper row) and spin (lower row) susceptibility between TPSC3 (blue), TPSC5 (orange), TPSC from Ref. [90] (green) and D-TRILEX (purple) at different $U$ and $J = 0.25U$ at $\beta = 2$.

discussion above. In general TPSC5 seems to outperform TPSC3, even though both fulfill the internal consistency check in the small to intermediate coupling region. Notably, there is no significant improvement in the case of large Hund's couplings when compared to the approach put forward in Ref. [90].

### 3.3.2 Comparison to D-TRILEX - Self-energy and susceptibilities

In addition to the quality of the TPEV's we can also asses the quality of interacting charge and spin susceptibilities as well as the self-energy. In the following we will compare our results to D-TRILEX results provided by the authors of Ref. [19]. For this, we consider a model of the form of Fig. 1 where an additional hopping imbalance is introduced between the red and blue orbitals

$$H = -\sum_{\alpha,\langle i,j\rangle,s} t_\alpha c^\dagger_{\alpha,s,i} c_{\alpha,s,j} + H_{HK} \,, \tag{60}$$

with $t_{\text{red}} = 1$ and $t_{\text{blue}} = 0.75$. $U$ is in the following given in units of $t_{\text{red}}$, while $J$ is fixed to $0.25U$. To obtain smooth curves along the irreducible path without the need for interpolation we enlarge the momentum mesh to $120 \times 120$. Furthermore, we compare quantities obtained without analytical continuation to rule it out as a source of error.

First, let us consider the self-energy at the lowest Matsubara frequency (Fig. 4). Qualitatively, all TPSC variants agree with D-TRILEX at both interactions values considered. Quantitatively, the TPSC results are not fully agreeing with each other which is somewhat expected from the different formulations. We note that TPSC3 appears to be furthest away from the D-TRILEX self-energy for this specific model, while TPSC5 and Ref. [66] appear to be better agreeing.

An analogous picture emerges when comparing the self-energy calculated with the different methods at different momentum points as a function of Matsubara frequency, see Fig. 5. We observe qualitative agreement but quantitative deviations. Note that at high frequency the noise of the DMFT impurity solver utilized to obtain the starting point in D-TRILEX becomes observable, especially in the real part of the self-energy.

A similar picture emerges for the susceptibility, see Fig. 6: all TPSC variants over(under)-estimate the spin(charge) susceptibility in comparison to D-TRILEX, in agreement with what was reported in Ref. [95]. However, the overestimation is weaker for TPSC5 and stronger for TPSC3. Most notably, the charge susceptibility is even more strongly suppressed than in prior TPSC formulations. This disagreement between the TPSC variants is a consequence of the vertex having a different from, see App. B.1, which leads to problems in the absence of $J$, see Ref. [90] but better agreement to the reference data at large $J$, see Fig. 3.

## 4 Conclusions and outlook

In this paper we introduced two variants of a fully self-consistent multi-orbital TPSC approach dubbed TPSC3 and TPSC5. We analyzed the structure of the equations and showed analytically that in the limit of vanishing Hund's coupling the Ansatz equations result in the same TPEVs as for the single-orbital model, which is unphysical. Furthermore, we provided an explanation for this behavior in terms of the underlying Hartree-Fock decomposition and its connection to the variational Hartree-Fock approach, as well as the general criteria under which a static self-energy is expected to be sufficiently accurate. By considering the local Hilbert space we also provided an understanding of why the approach performs better at larger $J/U$. Notably, this understanding allows to assess the expected quality of the results from TPSC by inspecting the local spectrum. On a qualitative level, we found that the TPEVs from TPSC5 outperform the ones from TPSC3 when compared to DMFT. In the limiting case of $J = U/3$ the new variants do not outperform the conceptually simpler but approximate approach suggested in Ref. [90].

While TPSC itself cannot be improved much without fundamentally modifying the basic approach, the recently proposed combination of TPSC and DMFT [91, 95] not only partially resolves the issue of wrong TPEVs due to the Hartree-Fock decoupling but also ensures the correct handling of correlations at the zeroth-order level. Thus, in multi-orbital systems, the application of TPSC in combination with DMFT appears as the most promising route to correct for the flaws uncovered in the present work and in Ref. [91]. A major advantage of TPSC in this formulation is that no vertex needs to be extracted from the DMFT simulation, which offers a numerically much cheaper alternative to D-TRILEX and related approaches [14, 19].

## Acknowledgments

The authors are grateful to A. Razpopov, P. P. Stavropoulos, L. Klebl and G. Rohringer for valuable discussions and E. Stepanov for providing the reference data from D-TRILEX.

**Funding information** We acknowledge support by the Deutsche Forschungsgemeinschaft (DFG, German Research Foundation) for funding through project QUAST-FOR5249 - 449872909 (project TP4) and (project TP6). J.Y. and P.W. also acknowledge support from SNSF Grant No. 200021-196966.

## A Kadanoff-Baym formalism

In the following we derive the central equations for the multi-orbital TPSC approach, following Ref. [80, 90]. We will only consider objects with the same number of in- and out-going legs. Therefore, we will stick to the definition that the first half of the indices are always in-going legs, while the second half are outgoing legs. Furthermore, to keep the equations

more managable we will use numbers as a short hand for a collection of quantum numbers $1 \equiv (o_1, s_1, \boldsymbol{r}_1, \tau_1)$. A Kronecker-delta in numbers is defined as the product of the Kronecker deltas/Delta-distributions of the individual quantum numbers.

Our starting point is the Green's function generating functional [112]

$$\mathcal{G}[\phi] = -\ln\left(\left\langle T_\tau e^{-c^\dagger_{o_{3'},s_{3'}}(\tau_{3'})c_{o_{1'},s_{1'}}(\tau_{1'})\phi_{o_{1'},s_{1'},o_{3'},s_{3'}}(\tau_{1'},\tau_{3'})}\right\rangle_S\right) = -\ln\left(\langle T_\tau S[\phi]\rangle_S\right) = \ln(Z[\phi]), \quad \text{(A.1)}$$

where the expectation value over the sources is calculated with regard to the action of the full system $S$. Primed variables are summed over. The external source field $\phi$ has been introduced as a mathematical trick and at the end of the calculation it should be set to 0. The single-particle Green's function is obtained as the functional derivative of $\mathcal{G}$ with respect to $\phi$,

$$G^{s_1,s_3}_{o_1,o_3;\phi}(\tau_1,\tau_3) = \frac{\delta\mathcal{G}[\phi]}{\delta\phi^{s_1,s_3}_{o_1,o_3}(\tau_3,\tau_1)} = -\frac{\langle T_\tau S[\phi] c_{o_1,s_1}(\tau_1) c^\dagger_{o_3,s_3}(\tau_3)\rangle_S}{Z[\phi]}$$
$$= -\langle T_\tau c_{o_1,s_1}(\tau_1) c^\dagger_{o_3,s_3}(\tau_3)\rangle_\phi. \quad \text{(A.2)}$$

The subscript $\phi$ indicates that the expectation value is to be calculated with the source fields added. With this, we perform a Legendre transformation resulting in the vertex generating functional or Luttinger-Ward functional $\Phi[G]$. The first functional derivative of $\Phi[G]$ is the self-energy

$$\Sigma^{s_1,s_3}_{o_1,o_3;\phi}(\tau_1,\tau_3) = -\frac{\delta\Phi[G]}{\delta G^{s_3,s_1}_{o_3,o_1;\phi}(\tau_3,\tau_1)}. \quad \text{(A.3)}$$

In the next step, we derive the equation of motion and through that obtain the Dyson equation relating the dressed Green's function and the self-energy. Our starting point is a general Hamiltonian with local and static interactions

$$H = t^{s_1,s_3}_{o_1,o_3}(\boldsymbol{r}_1,\boldsymbol{r}_3) c^\dagger_{o_3,s_3}(\boldsymbol{r}_3) c_{o_1,s_1}(\boldsymbol{r}_1) + \frac{1}{2} U^{s_1,s_2,s_3,s_4}_{o_1,o_2,o_3,o_4} c^\dagger_{o_3,s_3}(\boldsymbol{r}) c^\dagger_{o_4,s_4}(\boldsymbol{r}) c_{o_2,s_2}(\boldsymbol{r}) c_{o_1,s_1}(\boldsymbol{r}). \quad \text{(A.4)}$$

In this case, the Heisenberg equation of motion (in imaginary time) reads

$$\partial_{\tau_5} c_{o_5,s_5}(\boldsymbol{r}_5,\tau_5) = \left[H, c_{o_5,s_5}(\boldsymbol{r}_5,\tau_5)\right] \quad \text{(A.5)}$$

$$= t^{s_{1'},s_{3'}}_{o_{1'},o_{3'}}(\boldsymbol{r}_1',\boldsymbol{r}_3')\left[c^\dagger_{o_{3'},s_{3'}}(\boldsymbol{r}_3')c_{o_{1'},s_{1'}}(\boldsymbol{r}_1'), c_{o_5,s_5}(\boldsymbol{r}_5)\right]_H \quad \text{(A.6)}$$

$$+ \frac{1}{2} U^{s_{1'},s_{2'},s_{3'},s_{4'}}_{o_{1'},o_{2'},o_{3'},o_{4'}}\left[c^\dagger_{o_3,s_3}(\boldsymbol{r}')c^\dagger_{o_4,s_4}(\boldsymbol{r}')c_{o_2,s_2}(\boldsymbol{r}')c_{o_1,s_1}(\boldsymbol{r}'), c_{o_5,s_5}(\boldsymbol{r}_5)\right]_H$$

$$= -t^{s_{1'},s_{3'}}_{o_{1'},o_{3'}}(\boldsymbol{r}_1',\boldsymbol{r}_3')\left(\left\{c^\dagger_{o_{3'},s_{3'}}(\boldsymbol{r}_3'), c_{o_5,s_5}(\boldsymbol{r}_5)\right\} c_{o_{1'},s_{1'}}(\boldsymbol{r}_1')\right)_H \quad \text{(A.7)}$$

$$+ \frac{1}{2} U^{s_{1'},s_{2'},s_{3'},s_{4'}}_{o_{1'},o_{2'},o_{3'},o_{4'}}\left(\left[c^\dagger_{o_{3'},s_{3'}}(\boldsymbol{r}')c^\dagger_{o_{4'},s_{4'}}(\boldsymbol{r}'), c_{o_5,s_5}(\boldsymbol{r}_5)\right] c_{o_{2'},s_{2'}}(\boldsymbol{r}')c_{o_{1'},s_{1'}}(\boldsymbol{r}')\right)_H$$

$$= -t^{s_{1'},s_{3'}}_{o_{1'},o_{3'}}(\boldsymbol{r}_1',\boldsymbol{r}_3')c_{o_{1'},s_{1'}}(\boldsymbol{r}_1',\tau_5)\delta_{o_{3'},o_5}\delta_{s_{3'},s_5}\delta_{\boldsymbol{r}_3',\boldsymbol{r}_5} \quad \text{(A.8)}$$

$$- \frac{1}{2} U^{s_{1'},s_{2'},s_{3'},s_{4'}}_{o_{1'},o_{2'},o_{3'},o_{4'}}\left(\left(c^\dagger_{o_{3'},s_{3'}}(\boldsymbol{r}')\delta_{o_{4'},o_5}\delta_{s_{4'},s_5}\delta_{\boldsymbol{r}',\boldsymbol{r}_5} - \delta_{o_{3'},o_5}\delta_{s_{3'},s_5}\delta_{\boldsymbol{r}',\boldsymbol{r}_5}c^\dagger_{o_{4'},s_{4'}}(\boldsymbol{r}')\right)\right.$$
$$\left.\times c_{o_{2'},s_{2'}}(\boldsymbol{r}')c_{o_{1'},s_{1'}}(\boldsymbol{r}')\right)_H$$

$$= -t^{s_{1'},s_5}_{o_{1'},o_5}(\boldsymbol{r}_1',\boldsymbol{r}_5)c_{o_{1'},s_{1'}}(\boldsymbol{r}_1',\tau_5) \quad \text{(A.9)}$$

$$+ \frac{1}{2}\underbrace{\left(U^{s_{1'},s_{2'},s_{3'},s_5}_{o_{1'},o_{2'},o_{3'},o_5} - U^{s_{1'},s_{2'},s_5,s_{3'}}_{o_{1'},o_{2'},o_5,o_{3'}}\right)}_{=\Gamma^{s_{1'},s_{2'},s_{3'},s_5}_{o_{1'},o_{2'},o_{3'},o_5}}c^\dagger_{o_{3'},s_{3'}}(\boldsymbol{r}_5,\tau_5)c_{o_{2'},s_{2'}}(\boldsymbol{r}_5,\tau_5)c_{o_{1'},s_{1'}}(\boldsymbol{r}_5,\tau_5),$$

where we used that the Hamiltonian commutes with itself at any time to pull the time evolution operator out of the commutator. The time evolution operators are indicated by the subscript

$H$ on the respective outermost commutator or bracket. Furthermore, we remark that in the case of an anti-symmetrized interaction $U$ we have $\Gamma = \alpha U$, where the prefactor $\alpha$ depends on the exact definition of the anti-symmetrization.

In the next step, we derive the equation of motion of the Green's function, where we first have to perform a partial time ordering such that the derivative commutes with the time ordering operator (for brevity, we will use numbers as a shorthand for a collection of all quantum numbers, whenever they all do share the same index, i.e. $1 \equiv (o_1, s_1, \boldsymbol{r}_1, \tau_1)$),

$$-Z_\phi \partial_{\tau_1} G_{1,3} = \partial_{\tau_1} \left( \langle T_\tau S[\phi] c_1 c_3^\dagger \rangle \Theta(\tau_1 - \tau_3) - \langle T_\tau S[\phi] c_3^\dagger c_1 \rangle \Theta(\tau_3 - \tau_1) \right) \tag{A.10}$$

$$= \langle \partial_{\tau_1} T_\tau S[\phi] c_1 c_3^\dagger \rangle \Theta(\tau_1 - \tau_3) - \langle \partial_{\tau_1} T_\tau S[\phi] c_3^\dagger c_1 \rangle \Theta(\tau_3 - \tau_1) \tag{A.11}$$

$$+ \underbrace{\langle T_\tau (c_1 c_3^\dagger + c_3^\dagger c_1) S[\phi] \rangle \delta(\tau_1 - \tau_3)}_{= \delta_{1,3} Z[\phi]} .$$

To evaluate the time derivative we first have to time order the exponential and creation/annihilation operators partially, leaving out the time ordering w.r.t the argument of the annihilation operator of the exponential. We perform this fist for the first contribution

$$\langle \partial_{\tau_1} T_\tau S[\phi] c_1 c_3^\dagger \rangle = \left\langle T_\tau \partial_{\tau_1} e^{-\int_{\tau_1}^\beta d\tau_{3'} c_3^\dagger \phi_{1';3'} c_{1'}} c_1 e^{-\int_0^{\tau_1} d\tau_3 c_{3'}^\dagger \phi_{1';3'} c_{1'}} c_3^\dagger \right\rangle \tag{A.12}$$

$$= \left\langle T_\tau e^{-\int_{\tau_1}^\beta d\tau_{3'} c_3^\dagger \phi_{1';3'} c_{1'}} \left( \partial_{\tau_1} c_1 \right) e^{-\int_0^{\tau_1} d\tau_3 c_{3'}^\dagger \phi_{1';3'} c_{1'}} c_3^\dagger \right\rangle \tag{A.13}$$

$$+ \left\langle T_\tau e^{-\int_{\tau_1}^\beta d\tau_{3'} c_{3'}^\dagger \phi_{1';3'} c_{1'}} \delta_{\tau_{3'}, \tau_1} \left[ c_{3'}^\dagger \phi_{1';3'} c_{1'}, c_1 \right] e^{-\int_0^{\tau_1} d\tau_3 c_{3'}^\dagger \phi_{1';3'} c_{1'}} c_3^\dagger \right\rangle$$

$$= \langle T_\tau S[\phi] \left( \partial_{\tau_1} c_1 \right) c_3^\dagger \rangle - \langle T_\tau S[\phi] \phi_{1',1} c_{1'} c_3^\dagger \rangle . \tag{A.14}$$

The second contribution follows analogously and both of them can be recombined by utilizing the time ordering. From this we directly find

$$\partial_{\tau_1} G_{1,3} = -\frac{1}{Z_\phi} \langle T_\tau S[\phi] \left( \partial_{\tau_1} c_1 \right) c_3^\dagger \rangle - \phi_{1',1} G_{1',3} - \delta_{1,3} . \tag{A.15}$$

Now, let us insert the equation of motion for the annihilation operator,

$$\partial_{\tau_1} G_{1,3} = -\Gamma_{o_{1'},o_{2'},o_{3'},o_1}^{s_{1'},s_{2'},s_{3'},s_1} \langle T_\tau c_{o_{3'},s_{3'}}^\dagger(\tau_1) c_{o_{2'} s_{2'}}(\tau_1) c_{o_{1'},s_{1'}}(\tau_1) c_{o_3,s_3}^\dagger(\tau_3) \rangle_\phi \tag{A.16}$$

$$- t_{1',1} \delta_{\tau_1', \tau_1} G_{1',3} - \phi_{1',1} G_{1',3} - \delta_{1,3} . \tag{A.17}$$

By setting the interaction and the sources to zero, we identify the non-interacting Green's function as

$$(G^{(0)})_{1,3}^{-1} = \left( -\partial_{\tau_1} \delta_{1,3} - t_{1,3} \delta_{\tau_1, \tau_3} \right) . \tag{A.18}$$

Furthermore, we identify the self-energy as

$$\Sigma_{o_1,o_{1'}}^{s_1,s_{1'}}(\tau_1, \tau_1') G_{o_{1'},o_3}^{s_{1'},s_3}(\tau_1', \tau_3) = \Gamma_{o_{4'},o_{2'},o_{3'},o_1}^{s_{4'},s_{2'},s_{3'},s_1} \langle T_\tau c_{o_{3'},s_{3'}}^\dagger(\tau_1) c_{o_{2'} s_{2'}}(\tau_1) c_{o_{4'},s_{4'}}(\tau_1) c_{o_3,s_3}^\dagger(\tau_3) \rangle_\phi , \tag{A.19}$$

such that we arrive at the following Dyson equation for the interacting Green's function:

$$G_{1,3}^{-1} = (G^{(0)})_{1,3}^{-1} - \phi_{3,1} - \Sigma_{1,3} . \tag{A.20}$$

Note that due to the definition of the source-field term, there is an index swap in this equation compared to all other quantities appearing on the right hand side of the equation.

The two-particle expectation values on the RHS of Eq. (A.19) are directly linked to the generalized two-particle susceptibility, which is defined as the second functional derivative of $\mathcal{G}[\phi]$:

$$
\begin{aligned}
\chi^{s_1..s_4}_{o_1..o_4}(\tau_1..\tau_4) &= \frac{\delta G^{s_1,s_3}_{o_1,o_3;\phi}(\tau_1\tau_3)}{\delta\phi^{s_2,s_4}_{o_2,o_4;\phi}(\tau_2\tau_4)} \\
&= \langle T_\tau c^\dagger_{o_3,s_3}(\tau_3)c^\dagger_{o_4,s_4}(\tau_4)c_{o_2,s_2}(\tau_2)c_{o_1,s_1}(\tau_1)\rangle_\phi \\
&\quad - \langle T_\tau c_{o_1,s_1}(\tau_1)c^\dagger_{o_3,s_3}(\tau_3)\rangle_\phi \langle T_\tau c_{o_2,s_2}(\tau_2)c^\dagger_{o_4,s_4}(\tau_4)\rangle_\phi ,
\end{aligned}
\tag{A.21}
$$

while the two-particle vertex is given by

$$
\Gamma^{s_1..s_4}_{o_1..o_4}(\tau_1..\tau_4) = \frac{\delta\Sigma^{s_1,s_4}_{o_1,s_4}(\tau_1,\tau_4)}{\delta G^{s_3,s_2}_{o_3,s_2}(\tau_3,\tau_2)}.
\tag{A.22}
$$

With these definitions we obtain the Bethe-Salpeter equation connecting the interacting and non-interacting susceptibilities:

$$
0 = \frac{\delta\left(G^{-1}_{1,3'}G_{3',3}\right)}{\delta\phi_{2,4}}
\tag{A.23}
$$

$$
= G^{-1}_{1,3'}\frac{\delta(G_{3',3})}{\delta\phi_{2,4}} + \frac{\delta\left(G^{-1;0}_{1,3'}-\phi_{3',1}-\Sigma_{1,3'}\right)}{\delta\phi_{2,4}}G_{3',3}
\tag{A.24}
$$

$$
\Leftrightarrow \chi_{1,2,3,4} = G_{1,1'}\frac{\delta\left(\phi_{3',1'}+\Sigma_{1',3'}\right)}{\delta\phi_{2,4}}G_{3',3}
\tag{A.25}
$$

$$
= G_{1,4}G_{2,3} + G_{1,1'}\frac{\delta\Sigma_{1',3'}}{\delta\phi_{2,4}}G_{3',3}
\tag{A.26}
$$

$$
= G_{1,4}G_{2,3} + G_{1,1'}\frac{\delta\Sigma_{1',3'}}{\delta G_{2',4'}}\frac{\delta G_{2',4'}}{\delta\phi_{2,4}}G_{3',3}
\tag{A.27}
$$

$$
= G_{1,4}G_{2,3} + G_{1,1'}G_{3',3}\Gamma_{1',3',4',2'}\chi_{2',2,4',4}
\tag{A.28}
$$

$$
= \chi^0_{1,2,3,4} + \chi^0_{1,3',3,1'}\Gamma_{1',3',4',2'}\chi_{2',2,4',4}
\tag{A.29}
$$

$$
= \chi^0_{1,2,3,4} + \chi_{1,3',3,1'}\Gamma_{1',3',4',2'}\chi^0_{2',2,4',4}.
\tag{A.30}
$$

For numerical purposes it is often beneficial to reorder the indices such that the equations can be written as a matrix-matrix product.

In an $SU(2)$ symmetric system, we can furthermore use that all occurring quantities are diagonal in physical spin-space, thus resulting in spin-channel-specific equations which read (pulling out spin indices from the numbers)

$$
\chi^{S^{i'},S^{j'}}_{1,2,3,4} = \left(\chi^0_{1,2,3,4} + \chi^0_{1,3',3,1'}\Gamma_{1',3',4',2'}\chi_{2',2,4',4}\right)\sigma^i_{s_1,s_3}\sigma^j_{s_2,s_4}
\tag{A.31}
$$

$$
\Leftrightarrow \chi^{S^z,S^z}_{1,2,3,4} = \chi^{0;S^z,S^z}_{1,2,3,4} + \chi^{0;S^z,S^z}_{1,3',3,1'}\Gamma^{S^z,S^z}_{1',3',4',2'}\chi^{S^z,S^z}_{2',2,4',4}
\tag{A.32}
$$

$$
\Leftrightarrow \chi^{S^0,S^0}_{1,2,3,4} = \chi^{0;S^0,S^0}_{1,2,3,4} - \chi^{0;S^0,S^0}_{1,3',3,1'}\Gamma^{S^0,S^0}_{1',3',4',2'}\chi^{S^0,S^0}_{2',2,4',4},
\tag{A.33}
$$

where $S_xS_x$, $S_yS_y$ and $S_zS_z$ are related by $SU(2)$ symmetry and $S^0$ indicates the case in which $\sigma^i = \sigma^0_{s_1,s_2} = \delta_{s_1,s_2}$.

## A.1 Self-energy

In the following we will derive the equations required for the level-1 self-energy evaluation. For the following calculations, we assume an anti-symmetrized interaction tensor

$$\Sigma_{o_1,o_3}^{s_1,s_3}(\tau_1,\tau_3) = U_{o_{4'},o_{2'},o_{3'},o_1}^{s_{4'},s_{2'},s_{3'},s_1} \langle T_\tau c_{o_{3'},s_3}^\dagger(\tau_1)c_{o_5',s_5'}^\dagger(\tau_5')c_{o_{2'}s_{2'}}(\tau_1)c_{o_{4'},s_{4'}}(\tau_1)\rangle_\phi \tag{A.34}$$

$$\times (G^{-1})_{o_5',o_3}^{s_5',s_3}(\tau_5',\tau_3)$$

$$= U_{o_{4'},o_{2'},o_{3'},o_1}^{s_{4'},s_{2'},s_{3'},s_1}\Big(\chi_{o_{4'},o_{2'},o_{3'},o_5'}^{s_{4'},s_{2'},s_{3'},s_5'}(\tau_1,\tau_1,\tau_1,\tau_{5'}) \tag{A.35}$$

$$-G_{o_{4'},o_{3'}}^{s_{4'},s_{3'}}(\tau_1,\tau_1)G_{o_{2'},o_5'}^{s_{2'},s_5'}(\tau_1,\tau_5')\Big)(G^{-1})_{o_5',o_3}^{s_5',s_3}(\tau_5',\tau_3)$$

$$= U_{o_{4'},o_{2'},o_{3'},o_1}^{s_{4'},s_{2'},s_{3'},s_1}\Big(\chi_{o_{4'},o_{c'},o_{3'},o_{a'}}^{s_{4'},s_{c'},s_{3'},s_{a'}}(\tau_1,\tau_{c'},\tau_1,\tau_{a'})\Gamma_{o_{a'},o_{c'},o_{d'},o_{b'}}^{s_{a'},s_{c'},s_{d'},s_{b'}}(\tau_{a'},\tau_{c'},\tau_{d'},\tau_{b'})$$

$$\times G_{o_{2'},o_{d'}}^{s_{2'},s_{d'}}(\tau_1,\tau_{d'})\delta_{o_{b'},o_3}\delta_{s_{b'},s_3}\delta_{\tau_{b'},\tau_3} \tag{A.36}$$

$$+ \delta_{o_{4'},o_3}\delta_{s_{4'},s_3}\delta_{\tau_1,\tau_3}G_{o_{2'},o_{3'}}^{s_{2'},s_{3'}}(\tau_1,\tau_1)$$

$$- G_{o_{4'},o_{3'}}^{s_{4'},s_{3'}}(\tau_1,\tau_1)\delta_{o_{2'},o_3}\delta_{s_{2'},s_3}\delta_{\tau_1,\tau_3}\Big)$$

$$= U_{o_{4'},o_{2'},o_{3'},o_1}^{s_{4'},s_{2'},s_{3'},s_1}\Big(\chi_{o_{4'},o_{c'},o_{3'},o_{a'}}^{s_{4'},s_{c'},s_{3'},s_{a'}}(\tau_1,\tau_{c'},\tau_1,\tau_{a'}) \tag{A.37}$$

$$\times \Gamma_{o_{a'},o_{c'},o_{d'},o_3}^{s_{a'},s_{c'},s_{d'},s_3}(\tau_{a'},\tau_{c'},\tau_{d'},\tau_3)G_{o_{2'},o_{d'}}^{s_{2'},s_{d'}}(\tau_1,\tau_{d'})$$

$$+ 2\delta_{o_{4'},o_3}\delta_{s_{4'},s_3}\delta_{\tau_1,\tau_3}G_{o_{2'},o_{3'}}^{s_{2'},s_{3'}}(\tau_1,\tau_1)\Big).$$

Here, the latter term corresponds to the Hartree-Fock selfenergy, while the former corresponds to the Schwinger-Dyson equation.

## A.2 Zeroth order self-energy

Within TPSC the Luttinger Ward functional for an $SU(2)$ symmetric model with local initial interactions is approximated as

$$\Phi[G] = \frac{1}{2}G_{o_{3'},o_{1'}}(\tau_1',\tau_1')(2\Gamma_{o_{1'}o_{2'}o_{3'}o_{4'}} - \Gamma_{o_{1'}o_{2'}o_{4'}o_{3'}})G_{o_{4'},o_{2'}}(\tau_1',\tau_1'). \tag{A.38}$$

For a diagrammatic representation of the Luttinger-Ward functional within the TPSC approximation we refer the reader to Ref. [80] as such a representation is identical between single and multi-orbital. As discussed elsewhere [80,90,91] such a form of the Luttinger-Ward functional induces a local and static self-energy, which has to be accounted for in our calculations. We recall the definition of the self energy

$$\Sigma_{o_1,o_3}(\tau_1,\tau_3) = \frac{\delta\Phi[G]}{\delta G_{o_3,o_1}(\tau_3,\tau_1)} = \big(2\Gamma_{o_1,o_{2'},o_3,o_{4'}} - \Gamma_{o_1,o_{2'},o_{4'},o_3}\big)G_{o_{4'},o_{2'}}(\tau_1',\tau_1')\delta_{\tau_1,\tau_3}. \tag{A.39}$$

Notably, for general Hubbard-Kanamori parameters, the shifts induced by such a self-energy term are nontrivial and cannot be absorbed into the chemical potential, in contrast to the single band case. Thus, the zeroth order self-energy corrects the non-interacting Green's function and needs to be included in our calculations.

# B   Alternative derivation starting from Hubbard-Kanamori

In this Appendix, we derive the central equations for TPSC assuming that the bare Hamiltonian contains a Hubbard-Kanamori interaction and is $SU(2)$ invariant. In this special case, we start

from a simplified equation of motion derived by Zantout et al. [113] (see Eq. (5.323))

$$
\begin{aligned}
\Sigma^{s,s}_{o_1,o_{1'}}(\tau_1,\tau'_3)G^{s,s}_{o_{1'},o_3}(\tau'_3,\tau_3) = & -\sum_{o_4} U_{o_4,o_1}\langle n_{o_4,\bar{s}}(\tau_1)c_{o_1,s}(\tau_1)c^{\dagger}_{o_3,s}(\tau_3)\rangle \\
& -\sum_{o_4\neq o_1}\left(U_{o_4,o_1}-J_{o_4,o_1}\right)\langle n_{o_4,s}(\tau_1)c_{o_1,s}(\tau_1)c^{\dagger}_{o_3,s}(\tau_3)\rangle \\
& +\sum_{o_4\neq o_1} J_{o_4,o_1}\left(\langle c^{\dagger}_{o_4,\bar{s}}(\tau_1)c_{o_4,s}(\tau_1)c_{o_1,\bar{s}}(\tau_1)c^{\dagger}_{o_3,s}(\tau_3)\rangle\right) \\
& +\sum_{o_4\neq o_1} J_{o_4,o_1}\left(\langle c^{\dagger}_{o_1,\bar{s}}(\tau_1)c_{o_4,s}(\tau_1)c_{o_4,\bar{s}}(\tau_1)c^{\dagger}_{o_3,s}(\tau_3)\rangle\right).
\end{aligned}
\tag{B.1}
$$

In the derivation in the main text, the first two terms are encompassed in $D$, the third one in $C$ and the fourth one in $P$. We proceed again by performing a Hartree-Fock decompsition:

$$
\begin{aligned}
\Sigma^{s,s}_{o_1,o_{1'}}(\tau_1,\tau'_3)G^{s,s}_{o_{1'},o_3}(\tau'_3,\tau_3) \approx & \sum_{o_4} U_{o_4,o_1} G^{\bar{s},\bar{s}}_{o_4,o_4}(\tau_1,\tau_1)G^{s,s}_{o_1,o_3}(\tau_1,\tau_3) \\
& +\sum_{o_4\neq o_1}\left(U_{o_4,o_1}-J_{o_4,o_1}\right)\Big(G^{s,s}_{o_4,o_4}(\tau_1,\tau_1)G^{s,s}_{o_1,o_3}(\tau_1,\tau_3) \\
& \qquad\qquad\qquad -G^{s,s}_{o_1,o_4}(\tau_1,\tau_1)G^{s,s}_{o_4,o_3}(\tau_1,\tau_3)\Big) \\
& +\sum_{o_4\neq o_1} J_{o_4,o_1} G^{\bar{s},\bar{s}}_{o_1,o_4}(\tau_1,\tau_1)G^{s,s}_{o_4,o_3}(\tau_1,\tau_3) \\
& +\sum_{o_4\neq o_1} J_{o_4,o_1} G^{\bar{s},\bar{s}}_{o_4,o_1}(\tau_1,\tau_1)G^{s,s}_{o_4,o_3}(\tau_1,\tau_3),
\end{aligned}
\tag{B.2}
$$

where we introduced the Green's function as $G^{s,s'}_{o_1,o_3}(\tau_1,\tau_3)=\langle c^{\dagger}_{o_3,s'}(\tau_3)c_{o_1,s}(\tau_1)\rangle$. The next step is to introduce Ansätze such that the equal-time, equal-position limit is exactly recovered ($o_1=o_2$ and $\tau_1=\tau_2$). Here we have the freedom to regroup terms for the Ansätze either such that the Green's-functions do match (which is the form written above), or such that the prefactors do match. The former corresponds to TPSC5, the latter corresponds to TPSC3. As a reminder, we use the same short hand notation for redundant indices we introduced in the main text, i.e. $n^{s_1,s_2}_{o_1,o_2}=c^{\dagger}_{o_2,s_2}(\tau,\boldsymbol{r})c_{o_1,s_1}(\tau,\boldsymbol{r})$ and $n^{s_1}_{o_1}=n^{s_1,s_1}_{o_1,o_1}=c^{\dagger}_{o_1,s_1}(\tau,\boldsymbol{r})c_{o_1,s_1}(\tau,\boldsymbol{r})$

In the first case, we introduce

$$
\begin{aligned}
\Sigma^{s,s}_{o_1,o_{1'}}(\tau_1,\tau'_3)G^{s,s}_{o_{1'},o_2}(\tau'_3,\tau_2) \approx & -\sum_{o_4} \tilde{\alpha}_{o_4,o_1} G^{\bar{s},\bar{s}}_{o_4,o_4}(\tau_1,\tau_1)G^{s,s}_{o_1,o_2}(\tau_1,\tau_2) \\
& -\sum_{o_4\neq o_1} \tilde{\beta}_{o_4,o_1}\Big(G^{s,s}_{o_4,o_4}(\tau_1,\tau_1)G^{s,s}_{o_1,o_2}(\tau_1,\tau_2) \\
& \qquad\qquad\qquad -G^{s,s}_{o_1,o_4}(\tau_1,\tau_1)G^{s,s}_{o_4,o_2}(\tau_1,\tau_2)\Big) \\
& +\sum_{o_4\neq o_1} \tilde{\gamma}_{o_4,o_1} G^{\bar{s},\bar{s}}_{o_1,o_4}(\tau_1,\tau_1)G^{s,s}_{o_4,o_2}(\tau_1,\tau_2) \\
& +\sum_{o_4\neq o_1} \tilde{\delta}_{o_4,o_1} G^{\bar{s},\bar{s}}_{o_4,o_1}(\tau_1,\tau_1)G^{s,s}_{o_4,o_2}(\tau_1,\tau_2),
\end{aligned}
\tag{B.3}
$$

where the Ansätze are defined as

$$
\tilde{\alpha}_{o_4,o_1} = U_{o_4,o_1}\frac{\langle n_{o_4,\bar{s}}(\tau_1)n_{o_1,s}(\tau_1)\rangle}{n^{\bar{s},\bar{s}}_{o_4,o_4}n^{s,s}_{o_1,o_1}},
\tag{B.4}
$$

$$
\tilde{\beta}_{o_4,o_1} = (U_{o_4,o_1}-J_{o_4,o_1})\frac{\langle n_{o_4,s}(\tau_1)n_{o_1,s}(\tau_1)\rangle}{n^{s,s}_{o_4,o_4}n^{s,s}_{o_1,o_1}-n^{s,s}_{o_1,o_4}n^{s,s}_{o_4,o_1}},
\tag{B.5}
$$

$$\tilde{\gamma}_{o_4,o_1} = J_{o_4,o_1} \frac{\langle c^\dagger_{o_4,\bar{s}}(\tau_1) c_{o_4,s}(\tau_1) c_{o_1,\bar{s}}(\tau_1) c^\dagger_{o_1,s}(\tau_1)\rangle}{n^{\bar{s},\bar{s}}_{o_1,o_4} n^{s,s}_{o_4,o_1}}, \tag{B.6}$$

$$\tilde{\delta}_{o_4,o_1} = J_{o_4,o_1} \frac{\langle c^\dagger_{o_1,\bar{s}}(\tau_1) c_{o_4,s}(\tau_1) c_{o_4,\bar{s}}(\tau_1) c^\dagger_{o_2,s}(\tau_2)\rangle}{n^{\bar{s},\bar{s}}_{o_4,o_1} n^{s,s}_{o_4,o_1}}. \tag{B.7}$$

Here, we identify the Ansätze from TPSC5 by comparing the equations, i.e. $\tilde{D}^{\uparrow,\uparrow} = \tilde{\alpha}$, $\tilde{D}^{\uparrow,\downarrow} - \tilde{C}^{\uparrow,\downarrow} = \tilde{\beta}$, $\tilde{C}^{\uparrow,\uparrow} = \tilde{\gamma}$ and $\tilde{P} = \tilde{\delta}$.

For the second type of Ansätze, we first regroup all terms with $U$ as prefactor together and two of the $J$ terms such that we arrive at

$$
\begin{aligned}
\Sigma^{s,s}_{o_1,o_{1'}}(\tau_1,\tau'_3) G^{s,s}_{o_{1'},o_2}(\tau'_3,\tau_2) \approx -\sum_{o_4} \tilde{\epsilon}_{o_4,o_1} \Bigg( &\sum_{s'}(1-\delta_{o_1,o_4}\delta_{s',s}) G^{s',s'}_{o_4,o_4} G^{s,s}_{o_1,o_2}(\tau_1,\tau_2) \\
&-(1-\delta_{o_1,o_4}) G^{s,s}_{o_1,o_4} G^{s,s}_{o_4,o_2}(\tau_1,\tau_2)\Bigg) \\
+\sum_{o_4\neq o_1} \tilde{\chi}_{o_4,o_1} \Bigg( &\sum_{s'} G^{s',s'}_{o_1,o_4}(\tau_1,\tau_1) G^{s,s}_{o_4,o_2}(\tau_1,\tau_2) \\
&-G^{s,s}_{o_4,o_4}(\tau_1,\tau_1) G^{s,s}_{o_1,o_2}(\tau_1,\tau_2)\Bigg) \\
+\sum_{o_4\neq o_1} \tilde{\psi}_{o_4,o_1} &G^{\bar{s},\bar{s}}_{o_4,o_1}(\tau_1,\tau_1) G^{s,s}_{o_4,o_2}(\tau_1,\tau_2),
\end{aligned}
\tag{B.8}
$$

where the Ansätze are defined as

$$\tilde{\epsilon}_{o_4,o_1} = U_{o_4,o_1} \frac{\left(1-\delta_{o_1,o_4}\right)\langle n_{o_4,s}(\tau_1) n_{o_1,s}(\tau_1)\rangle + \langle n_{o_4,\bar{s}}(\tau_1) n_{o_1,s}(\tau_1)\rangle}{\sum_{s'} n^{s',s'}_{o_4,o_4} n^{s,s}_{o_1,o_1} - n^{s,s}_{o_1,o_4} n^{s,s}_{o_4,o_1}}, \tag{B.9}$$

$$\tilde{\chi}_{o_4,o_1} = \left(1-\delta_{o_1,o_4}\right) J_{o_4,o_1} \frac{\langle n_{o_4,\bar{s}}(\tau_1) n_{o_1,s}(\tau_1)\rangle + \langle c^\dagger_{o_4,\bar{s}}(\tau_1) c_{o_4,s}(\tau_1) c_{o_1,\bar{s}}(\tau_1) c^\dagger_{o_1,s}(\tau_1)\rangle}{\sum_{s'} n^{s',s'}_{o_1,o_4} n^{s,s}_{o_4,o_1} - n^{s,s}_{o_4,o_4} n^{s,s}_{o_1,o_1}}, \tag{B.10}$$

$$\tilde{\psi}_{o_4,o_1} = J_{o_4,o_1} \frac{\langle c^\dagger_{o_1,\bar{s}}(\tau_1) c_{o_4,s}(\tau_1) c_{o_4,\bar{s}}(\tau_1) c^\dagger_{o_2,s}(\tau_2)\rangle}{n^{\bar{s},\bar{s}}_{o_4,o_1} n^{s,s}_{o_4,o_1}}. \tag{B.11}$$

Here we can identify the Ansätze from TPSC3 by $\tilde{\epsilon} = \tilde{D}$, $\tilde{\chi} = \tilde{C}$ and $\tilde{\psi} = \tilde{C}$.

## B.1 Reproducing results from Ref. [90] and Ref. [89]

With the explicit re-derivation along the same lines as in Ref. [90], we find that to reproduce their Ansatz equations, we have to take the equations derived for TPSC5 and fix both $\tilde{C}^{\uparrow,\downarrow}$ and $\tilde{P}$ to the non-renormalized Hund's coupling. Furthermore, the interaction has to be constrained to

$$\Gamma^{S^3 S^3}_{o_1 o_2 o_3 o_4} = \delta_{o_1,o_3}\delta_{o_2,o_4}\left(\tilde{D}^{\downarrow\uparrow}_{o_4,o_1} - \tilde{D}^{\uparrow\uparrow}_{o_4,o_1} + \tilde{C}^{\uparrow\uparrow}_{o_4,o_1}\right) + \tilde{C}^{\uparrow\downarrow}_{o_1,o_3}\delta_{o_3,o_2}\delta_{o_1,o_4} + \tilde{P}_{o_3,o_1}\delta_{o_1,o_2}\delta_{o_3,o_4}, \tag{B.12}$$

i.e., we need to neglect the contribution of the same spin terms to the orbital combination $\delta_{o_3,o_2}\delta_{o_1,o_4}$.

Comparing our approach to Ref. [89], The vertex differs in the charge channel as the Hund's coupling is neglected. Additionally, as in Ref. [90] no renormalization for the Hund's coupling terms is introduced. Thus again we have to constrain the interaction accordingly. In both cases, the utilized sum rules are a subset of the ones we employ.

# C Derivation of the interaction from functional derivatives

In the following, we will derive the interaction vertices directly from the functional derivative of the self-energy. Since TPSC3 is formally a special case of TPSC5, we will stick with the spin dependent vertices and evaluate the special case in the end. Before starting, we note that formally, we can rewrite the Ansätze in equilibrium utilizing $SU(2)$-invariance as [80]

$$\tilde{D}_{o_1,o_4}^{s_1,s_4} = \frac{2D_{o_1,o_4}\left(\langle n_{o_4}^{s_4} n_{o_1}^{s_1}\rangle + \langle n_{o_4}^{\bar{s}_4} n_{o_1}^{\bar{s}_1}\rangle\right)}{\left(\langle n_{o_4}^{s_4}\rangle + \langle n_{o_4}^{\bar{s}_4}\rangle\right)\left(\langle n_{o_1}^{s_1}\rangle + \langle n_{o_1}^{\bar{s}_1}\rangle\right) - \left(\langle n_{o_1,o_4}^{s_1,s_4}\rangle + \langle n_{o_1,o_4}^{\bar{s}_1,\bar{s}_4}\rangle\right)\left(\langle n_{o_4,o_1}^{s_4,s_1}\rangle + \langle n_{o_4,o_1}^{\bar{s}_4,\bar{s}_1}\rangle\right)},$$

$$\tilde{C}_{o_1,o_4}^{s_1,s_4} = \frac{2C_{o_1,o_4}\left(\langle n_{o_4}^{s_1,s_4} n_{o_1}^{s_4,s_1}\rangle + \langle n_{o_4}^{\bar{s}_1,\bar{s}_4} n_{o_1}^{\bar{s}_4,\bar{s}_1}\rangle\right)}{\left(\langle n_{o_4}^{s_1,s_4}\rangle + \langle n_{o_4}^{\bar{s}_1,\bar{s}_4}\rangle\right)\left(\langle n_{o_1}^{s_4,s_1}\rangle + \langle n_{o_1}^{\bar{s}_4,\bar{s}_1}\rangle\right) - \left(\langle n_{o_1,o_4}^{s_4}\rangle + \langle n_{o_1,o_4}^{\bar{s}_4}\rangle\right)\left(\langle n_{o_4,o_1}^{s_1}\rangle + \langle n_{o_4,o_1}^{\bar{s}_1}\rangle\right)},$$

$$\tilde{P}_{o_1,o_4}^{s_1} = P_{o_1,o_4}\frac{\langle n_{o_4 o_1}^{s_1,\bar{s}_1} n_{o_4,o_1}^{\bar{s}_1,s_1}\rangle}{\langle n_{o_4 o_1}^{s_1,\bar{s}_1}\rangle\langle n_{o_4 o_1}^{\bar{s}_1,s_1}\rangle - \langle n_{o_4,o_1}^{\bar{s}_1}\rangle\langle n_{o_4,o_1}^{s_1}\rangle}.$$

By rewriting the Ansätze like this, we make it transparent that they are symmetric functional of $G^\sigma$ under the exchange of $G^\uparrow$ and $G^\downarrow$. Thereby, we explicitly find that

$$\frac{\delta X^{\sigma,\sigma'}}{\delta G^{\sigma''}} = \frac{\delta X^{\sigma,\sigma'}}{\delta G^{\bar{\sigma}''}}, \tag{C.1}$$

holds in the limit of zero external field, which is the limit we are ultimately interested in. This we will use later to argue that all terms proportional to functional derivatives of the Ansätze do cancel.

The starting point for the derivation is the definition of the self-energy in terms of the Ansätze

$$\begin{aligned}\Sigma_{o_1,o_3}^{s_1,s_3}(\tau_1;\tau_3) = \delta_{\tau_1,\tau_3}\Big(&\tilde{D}_{o_4',o_1}^{s_4',s_1} G_{o_4',o_4'}^{s_4',s_4'}(\tau_1;\tau_1)\delta_{s_1,s_3}\delta_{o_1,o_3} - \tilde{D}_{o_3,o_1}^{s_3,s_1} G_{o_1,o_3}^{s_1,s_3}(\tau_1;\tau_1)\\ &+ \tilde{C}_{o_1,o_3}^{s_1,s_3'} G_{o_1,o_3}^{s_3',s_3'}(\tau_1;\tau_1)\delta_{s_1,s_3} - \tilde{C}_{o_1,o_3'}^{s_1,s_3} G_{o_3',o_3'}^{s_1,s_3}(\tau_1;\tau_1)\delta_{o_1,o_3}\\ &+ \tilde{P}_{o_3,o_1}^{s_1,\bar{s}_1} G_{o_3,o_1}^{\bar{s}_1,\bar{s}_1}(\tau_1;\tau_1)\delta_{s_1,s_3} - \tilde{P}_{o_3,o_1}^{s_1,\bar{s}_1} G_{o_3,o_1}^{s_1,\bar{s}_1}(\tau_1;\tau_1)\delta_{\bar{s}_1,s_3}\Big).\end{aligned} \tag{C.2}$$

The vertex in Pauli-matrix spin space is given as

$$\Gamma_{o_1 o_2 o_3 o_4}^{S^i S^j}(\tau_1,\tau_2,\tau_4,\tau_1) = \frac{1}{2}\sum_{s_1,s_2,s_3,s_4}\sigma_{s_1,s_4}^i \frac{\delta\Sigma_{o_1,o_4}^{s_1,s_4}(\tau_1,\tau_1)}{\delta G_{o_3,o_2}^{s_3,s_2}(\tau_3,\tau_2)}\sigma_{\bar{s}_2,\bar{s}_3}^j. \tag{C.3}$$

Before the explicit calculation, we recapitulate that all Ansätze obey the following symmetries:

$$X^{\uparrow,\uparrow} = X^{\downarrow,\downarrow} \quad \text{and} \quad X^{\downarrow,\uparrow} = X^{\uparrow,\downarrow}. \tag{C.4}$$

Furthermore, they are Hermitian ($X^\dagger = X$) and real. These properties further imply that the self-energy obeys $\Sigma^{s_1,s_2} = \Sigma\delta_{s_1,s_2}$. Therefore, we have only two distinct cases defined by different spin alignments in the denominator and numerator. First, let us focus on the case $S^i = S^j = S^3$, for which we have to evaluate

$$\Gamma_{o_1 o_2 o_3 o_4}^{S^i S^j}(\tau_1,\tau_2,\tau_4,\tau_1) = \frac{\delta\Sigma_{o_1,o_4}^{\downarrow,\downarrow}(\tau_1,\tau_1)}{\delta G_{o_3,o_2}^{\uparrow,\uparrow}(\tau_3,\tau_2)} - \frac{\delta\Sigma_{o_1,o_4}^{\uparrow,\uparrow}(\tau_1,\tau_1)}{\delta G_{o_3,o_2}^{\uparrow,\uparrow}(\tau_3,\tau_2)}. \tag{C.5}$$

To make the derivation more manageable, we perform it term by term, thus leaving us with six contributions from Eq. (C.2). The first contribution reads

$$-\delta_{o_1,o_4}\frac{\delta\left(\tilde{D}^{s_4',\uparrow}_{o_4',o_1}G^{s_4',s_4'}_{o_4',o_4'}(\tau_1;\tau_1)\right)}{\delta G^{\uparrow,\uparrow}_{o_3,o_2}(\tau_3,\tau_2)}+\delta_{o_1,o_4}\frac{\delta\left(\tilde{D}^{s_4',\downarrow}_{o_4',o_1}G^{s_4',s_4'}_{o_4',o_4'}(\tau_1;\tau_1)\right)}{\delta G^{\uparrow,\uparrow}_{o_3,o_2}(\tau_3,\tau_2)} \tag{C.6}$$

$$=-\delta_{o_1,o_4}\left[G^{s_4',s_4'}_{o_4',o_4'}(\tau_1;\tau_1)\left(\frac{\delta\tilde{D}^{s_4',\uparrow}_{o_4',o_1}}{\delta G^{\uparrow,\uparrow}_{o_3,o_2}(\tau_3,\tau_2)}-\frac{\delta\tilde{D}^{s_4',\downarrow}_{o_4',o_1}}{\delta G^{\uparrow,\uparrow}_{o_3,o_2}(\tau_3,\tau_2)}\right)\right. \tag{C.7}$$

$$\left.-\left(\tilde{D}^{s_4',\uparrow}_{o_4',o_1}-\tilde{D}^{s_4',\downarrow}_{o_4',o_1}\right)\frac{\delta G^{s_4',s_4'}_{o_4',o_4'}(\tau_1;\tau_1)}{\delta G^{\uparrow,\uparrow}_{o_3,o_2}(\tau_3,\tau_2)}\right]$$

$$=\delta_{\tau_1,\tau_2}\delta_{\tau_1,\tau_4}\delta_{o_1,o_4}\delta_{o_2,o_3}\left(\tilde{D}^{\uparrow,\downarrow}_{o_3,o_1}-\tilde{D}^{\uparrow,\uparrow}_{o_3,o_1}\right). \tag{C.8}$$

Here the first term in the second line vanishes due to $SU(2)$ symmetry. Note that if we would have spin-orbit coupling the time-reversal symmetry would not lead to a cancellation of this contribution, in contrast to the single-band case [87]. The second contribution reads

$$\frac{\delta\left(\tilde{D}^{\uparrow,\uparrow}_{o_4,o_1}G^{\uparrow,\uparrow}_{o_1,o_4}(\tau_1;\tau_1)\right)}{\delta G^{\uparrow,\uparrow}_{o_3,o_2}(\tau_3,\tau_2)}-\frac{\delta\left(\tilde{D}^{\downarrow,\downarrow}_{o_4,o_1}G^{\downarrow,\downarrow}_{o_1,o_4}(\tau_1;\tau_1)\right)}{\delta G^{\uparrow,\uparrow}_{o_3,o_2}(\tau_3,\tau_2)}$$

$$=\delta_{o_1,o_3}\delta_{o_2,o_4}\delta_{\tau_1,\tau_2}\delta_{\tau_1,\tau_4}\tilde{D}^{\uparrow,\uparrow}_{o_4,o_1}+G^{\uparrow,\uparrow}_{o_1,o_4}(\tau_1;\tau_1)\left(\frac{\delta\tilde{D}^{\uparrow,\uparrow}_{o_4,o_1}}{\delta G^{\uparrow,\uparrow}_{o_3,o_2}(\tau_3,\tau_2)}-\frac{\delta\tilde{D}^{\downarrow,\downarrow}_{o_4,o_1}}{\delta G^{\uparrow,\uparrow}_{o_3,o_2}(\tau_3,\tau_2)}\right) \tag{C.9}$$

$$=\delta_{\tau_1,\tau_2}\delta_{\tau_1,\tau_4}\delta_{o_1,o_3}\delta_{o_2,o_4}\tilde{D}^{\uparrow,\uparrow}_{o_4,o_1}. \tag{C.10}$$

The first terms proportional to $C$ reads

$$-\frac{\delta\left(\tilde{C}^{\uparrow,s_3'}_{o_1,o_4}G^{s_3',s_3'}_{o_1,o_4}(\tau_1;\tau_1)\right)}{\delta G^{\uparrow,\uparrow}_{o_3,o_2}(\tau_3,\tau_2)}+\frac{\delta\left(\tilde{C}^{\downarrow,s_3'}_{o_1,o_4}G^{s_3',s_3'}_{o_1,o_4}(\tau_1;\tau_1)\right)}{\delta G^{\uparrow,\uparrow}_{o_3,o_2}(\tau_3,\tau_2)}$$

$$=-\delta_{o_1,o_3}\delta_{o_2,o_4}\delta_{\tau_1,\tau_2}\delta_{\tau_1,\tau_4}\left(\tilde{C}^{\uparrow,\uparrow}_{o_1,o_4}-\tilde{C}^{\downarrow,\uparrow}_{o_1,o_4}\right)$$

$$-G^{s_4',s_4'}_{o_1,o_4}(\tau_1;\tau_1)\left(\frac{\delta\tilde{C}^{\uparrow,s_4'}_{o_1,o_4}}{\delta G^{\uparrow,\uparrow}_{o_3,o_2}(\tau_3,\tau_2)}-\frac{\delta\tilde{C}^{\downarrow,s_4'}_{o_1,o_4}}{\delta G^{\uparrow,\uparrow}_{o_3,o_2}(\tau_3,\tau_2)}\right) \tag{C.11}$$

$$=\delta_{\tau_1,\tau_2}\delta_{\tau_1,\tau_4}\delta_{o_1,o_3}\delta_{o_2,o_4}\left(\tilde{C}^{\downarrow,\uparrow}_{o_1,o_4}-\tilde{C}^{\uparrow,\uparrow}_{o_1,o_4}\right), \tag{C.12}$$

the second term

$$+\delta_{o_1,o_4}\frac{\delta\left(\tilde{C}^{\uparrow,\uparrow}_{o_1,o_3'}G^{\uparrow,\uparrow}_{o_3',o_3'}(\tau_1;\tau_1)\right)}{\delta G^{\uparrow,\uparrow}_{o_3,o_2}(\tau_3,\tau_2)}-\delta_{o_1,o_4}\frac{\delta\left(\tilde{C}^{\downarrow,\downarrow}_{o_1,o_3'}G^{\downarrow,\downarrow}_{o_3',o_3'}(\tau_1;\tau_1)\right)}{\delta G^{\uparrow,\uparrow}_{o_3,o_2}(\tau_3,\tau_2)}$$

$$=+\delta_{o_1,o_4}\delta_{o_2,o_3}\delta_{\tau_1,\tau_2}\delta_{\tau_1,\tau_4}\tilde{C}^{\uparrow,\uparrow}_{o_1,o_3}$$

$$+\delta_{o_1,o_4}G^{\downarrow,\downarrow}_{o_3',o_3'}(\tau_1;\tau_1)\left(\frac{\delta\tilde{C}^{\uparrow,\uparrow}_{o_1,o_3'}}{\delta G^{\uparrow,\uparrow}_{o_3,o_2}(\tau_3,\tau_2)}-\frac{\delta\tilde{C}^{\downarrow,\downarrow}_{o_1,o_3'}}{\delta G^{\uparrow,\uparrow}_{o_3,o_2}(\tau_3,\tau_2)}\right) \tag{C.13}$$

$$=\delta_{\tau_1,\tau_2}\delta_{\tau_1,\tau_4}\delta_{o_1,o_4}\delta_{o_2,o_3}\tilde{C}^{\uparrow,\uparrow}_{o_1,o_3}, \tag{C.14}$$

while for the terms proportional to the $P$ Ansatz we find (for this the second term drops out

due to the spin off-diagonal component of the Green's function being 0)

$$
-\left( \frac{\delta \tilde{P}^{\uparrow,\downarrow}_{o_4,o_1}}{\delta G^{\uparrow,\uparrow}_{o_3,o_2}(\tau_3,\tau_2)} - \frac{\delta \tilde{P}^{\downarrow,\uparrow}_{o_4,o_1}}{\delta G^{\uparrow,\uparrow}_{o_3,o_2}(\tau_3,\tau_2)} \right) \left( G^{\downarrow,\downarrow}_{o_4,o_1}(\tau_1;\tau_1) - G^{\uparrow,\uparrow}_{o_4,o_1}(\tau_1;\tau_1) \right)
$$
$$
-\left( \frac{\delta G^{\downarrow,\downarrow}_{o_4,o_1}(\tau_1;\tau_1)}{\delta G^{\uparrow,\uparrow}_{o_3,o_2}(\tau_3,\tau_2)} - \frac{G^{\uparrow,\uparrow}_{o_4,o_1}(\tau_1;\tau_1)}{\delta G^{\uparrow,\uparrow}_{o_3,o_2}(\tau_3,\tau_2)} \right) \tilde{P}^{\uparrow,\downarrow}_{o_4,o_1} = \delta_{\tau_1,\tau_2} \delta_{\tau_1,\tau_4} \delta_{o_1,o_2} \delta_{o_3,o_4} P^{\uparrow,\downarrow}_{o_3,o_1}. \tag{C.15}
$$

Putting everything together we find

$$
\Gamma^{S^3 S^3;5}_{o_1 o_2 o_3 o_4}(\tau_1,\tau_2,\tau_1,\tau_4) = \frac{\delta \Sigma^{\uparrow,\uparrow}_{o_1,o_3}(\tau_1,\tau_1)}{\delta G^{\uparrow,\uparrow}_{o_3,o_2}(\tau_3,\tau_2)} - \frac{\delta \Sigma^{\downarrow,\downarrow}_{o_1,o_3}(\tau_1,\tau_1)}{\delta G^{\uparrow,\uparrow}_{o_3,o_2}(\tau_3,\tau_2)} \tag{C.16}
$$
$$
= \delta_{\tau_1,\tau_2} \delta_{\tau_1,\tau_4} \Big( \delta_{o_1,o_4} \delta_{o_2,o_3} \big( \tilde{D}^{\uparrow,\downarrow}_{o_3,o_1} - \tilde{D}^{\uparrow,\uparrow}_{o_3,o_1} \big) + \delta_{o_1,o_3} \delta_{o_2,o_4} \tilde{D}^{\uparrow,\uparrow}_{o_4,o_1}
$$
$$
+ \delta_{o_1,o_3} \delta_{o_2,o_4} \big( \tilde{C}^{\downarrow,\uparrow}_{o_1,o_4} - \tilde{C}^{\uparrow,\uparrow}_{o_1,o_4} \big) + \delta_{o_1,o_4} \delta_{o_2,o_3} \tilde{C}^{\uparrow,\uparrow}_{o_1,o_3}
$$
$$
+ \delta_{o_1,o_2} \delta_{o_3,o_4} P^{\uparrow,\downarrow}_{o_3,o_1} \Big). \tag{C.17}
$$

To get the vertex for TPSC3 we have to set $\tilde{D}^{\uparrow\uparrow} = \tilde{D}^{\uparrow\downarrow}$ and $\tilde{C}^{\uparrow\uparrow} = \tilde{C}^{\uparrow\downarrow} = \tilde{C}$. Here, we have to keep in mind that only $\tilde{D}^{\uparrow\downarrow}$ contains contributions for equal orbitals.

$$
\Gamma^{S^3 S^3;3}_{o_1 o_2 o_3 o_4}(\tau_1,\tau_2,\tau_1,\tau_4) = \delta_{\tau_1,\tau_2} \delta_{\tau_1,\tau_4} \big( \delta_{o_1,o_4} \delta_{o_2,o_3} \tilde{D}_{o_3,o_1} + \delta_{o_1,o_4} \delta_{o_2,o_3} \tilde{C}_{o_1,o_3} + \delta_{o_1,o_2} \delta_{o_3,o_4} P_{o_3,o_1} \big). \tag{C.18}
$$

# D  Particle-hole symmetric Ansätze

In the following, we will explicitly derive the particle-hole symmetrized Ansatz equation. To this end we recapitulate the ansatz equations from TPSC5

$$
\tilde{D}^{s_1,s_4}_{o_1,o_4} = D_{o_1,o_4} \frac{\langle n^{s_4}_{o_4} n^{s_1}_{o_1} \rangle}{\langle n^{s_4}_{o_4} \rangle \langle n^{s_1}_{o_1} \rangle - \langle n^{s_1,s_4}_{o_1,o_4} \rangle \langle n^{s_4,s_1}_{o_4,o_1} \rangle}, \tag{D.1}
$$

$$
\tilde{C}^{s_1,s_4}_{o_1,o_4} = C_{o_1,o_4} \frac{\langle n^{s_1,s_4}_{o_4} n^{s_4,s_1}_{o_1} \rangle}{\langle n^{s_1,s_4}_{o_4} \rangle \langle n^{s_4,s_1}_{o_1} \rangle - \langle n^{s_4}_{o_1,o_4} \rangle \langle n^{s_1}_{o_4,o_1} \rangle}, \tag{D.2}
$$

$$
\tilde{P}^{s_1}_{o_1,o_4} = P_{o_1,o_4} \frac{\langle n^{s_1,\bar{s}_1}_{o_4 o_1} n^{\bar{s}_1,s_1}_{o_4,o_1} \rangle}{\langle n^{s_1,\bar{s}_1}_{o_4 o_1} \rangle \langle n^{\bar{s}_1,s_1}_{o_4 o_1} \rangle - \langle n^{\bar{s}_1}_{o_4,o_1} \rangle \langle n^{s_1}_{o_4,o_1} \rangle}. \tag{D.3}
$$

The symmetric version of the Ansatz equations of TPSC3 is obtained by performing the sum over $s_4$ in both the enumerator and the denominator.

To obtain a particle-hole symmetric set of Ansätze, we explicitly average the usual Ansatz with their particle-hole transformed counterpart $c^\dagger \to c$ and vice versa. This results in

$$
\tilde{D}^{s_1,s_4}_{o_1,o_4} = \frac{D_{o_1,o_4}}{2} \left( \frac{\langle n^{s_4}_{o_4} n^{s_1}_{o_1} \rangle}{\langle n^{s_4}_{o_4} \rangle \langle n^{s_1}_{o_1} \rangle - \langle n^{s_1,s_4}_{o_1,o_4} \rangle \langle n^{s_4,s_1}_{o_4,o_1} \rangle} \right. \tag{D.4}
$$
$$
\left. + \frac{\langle (1 - n^{s_4}_{o_4})(1 - n^{s_1}_{o_1}) \rangle}{\langle 1 - n^{s_4}_{o_4} \rangle \langle 1 - n^{s_1}_{o_1} \rangle - \langle \delta_{s_1,s_4} \delta_{o_1,o_4} - n^{s_1,s_4}_{o_1,o_4} \rangle \langle \delta_{s_1,s_4} \delta_{o_1,o_4} - n^{s_4,s_1}_{o_4,o_1} \rangle} \right),
$$

$$
\tilde{C}^{s_1,s_4}_{o_1,o_4} = \frac{C_{o_1,o_4}}{2} \left( \frac{\langle n^{s_1,s_4}_{o_4} n^{s_4,s_1}_{o_1} \rangle}{\langle n^{s_1,s_4}_{o_4} \rangle \langle n^{s_4,s_1}_{o_1} \rangle - \langle n^{s_4}_{o_1,o_4} \rangle \langle n^{s_1}_{o_4,o_1} \rangle} \right. \tag{D.5}
$$
$$
\left. + \frac{\langle (\delta_{s_1,s_4} - n^{s_1,s_4}_{o_4})(\delta_{s_1,s_4} - n^{s_4,s_1}_{o_1}) \rangle}{\langle \delta_{s_1,s_4} - n^{s_1,s_4}_{o_4} \rangle \langle \delta_{s_1,s_4} - n^{s_4,s_1}_{o_1} \rangle - \langle \delta_{o_1,o_4} - n^{s_4}_{o_1,o_4} \rangle \langle \delta_{o_1,o_4} - n^{s_1}_{o_4,o_1} \rangle} \right),
$$

$$\tilde{P}^{s_1}_{o_1,o_4} = \frac{P_{o_1,o_4}}{2} \left( \frac{\langle n^{s_1,\bar{s}_1}_{o_4 o_1} n^{\bar{s}_1,s_1}_{o_4,o_1} \rangle}{\langle n^{s_1,\bar{s}_1}_{o_4 o_1} \rangle \langle n^{\bar{s}_1,s_1}_{o_4 o_1} \rangle - \langle n^{\bar{s}_1}_{o_4,o_1} \rangle \langle n^{s_1}_{o_4,o_1} \rangle} \right.$$
$$\left. + \frac{\langle n^{s_1,\bar{s}_1}_{o_4 o_1} n^{\bar{s}_1,s_1}_{o_4,o_1} \rangle}{\langle n^{s_1,\bar{s}_1}_{o_4 o_1} \rangle \langle n^{\bar{s}_1,s_1}_{o_4 o_1} \rangle - \langle \delta_{o_1,o_4} - n^{\bar{s}_1}_{o_4,o_1} \rangle \langle \delta_{o_1,o_4} - n^{s_1}_{o_4,o_1} \rangle} \right). \tag{D.6}$$

These modified Ansatz equations should be used whenever one studies a particle-hole symmetric system.

# E Comparison to ED

In this appendix, we compare the results from TPSC with ED. Due to the restriction to finite-size systems in ED, this is essentially a comparison of Hartree-Fock with ED, since the fixing of the local double occupancies only gives minor corrections on top of Hartree-Fock. This comparison is complementary to the analytical considerations in Sec. 3.1. There, we traced back the breakdown of TPSC to the incorrect description of the local thermal expectation values by Hartree-Fock. To emphasize this, we show data at two temperatures $\beta = 0.5t, 2t$, see Fig. 8. Furthermore, by reducing the filling we also find better agreement with ED in the dimer case, see Fig. 7a, while adding a kinetic inter-orbital coupling at half filling does not improve the results, see Fig. 7b.

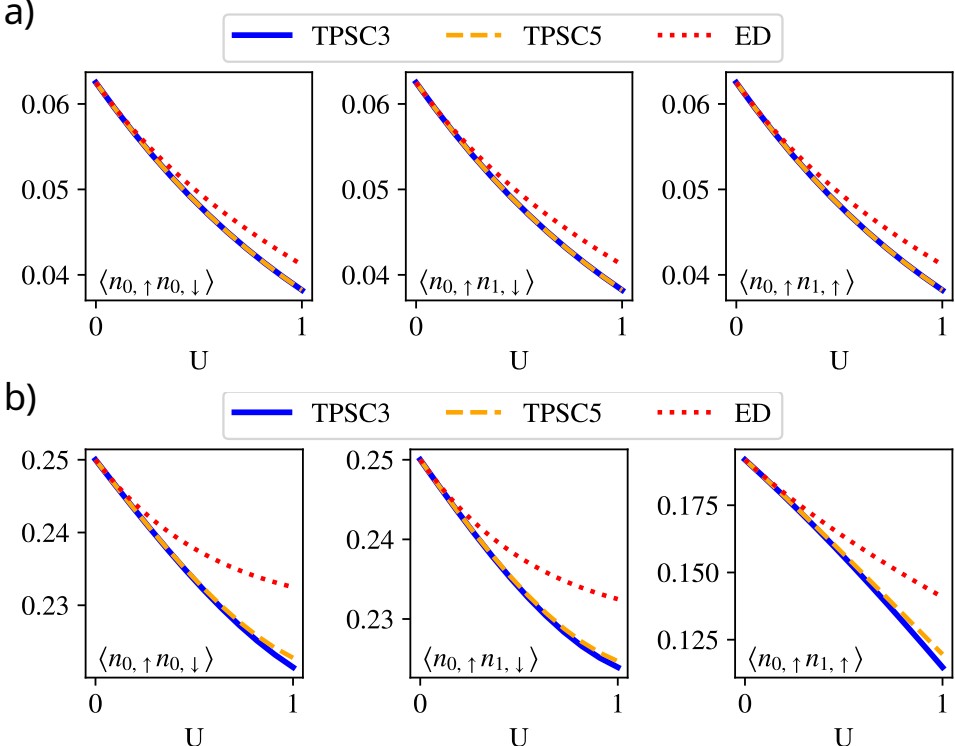

Figure 7: Comparison between TPSC and ED at $\beta = 2$, for different $U$ at $J = 0$ at quarter filling (panel a) and with a kinetic inter-orbital coupling of unit strength (panel b). For these comparisons we include the zeroth order self-energy correction.

The model we study is again akin to Fig. 1 and its Hamiltonian is given by

$$H = \sum_{\sigma} t \left( c^{\dagger}_{0,s'} c_{1,s'} + c^{\dagger}_{1,s'} c_{0,s'} \right) + H^{HK} , \qquad (E.1)$$

where we use the same Hubbard-Kanamori interaction as in the main text, see Eq. (47). As in the main text $U$ and $J$ are measured in units of $t$, which is set to 1. We consider the weak coupling limit in which TPSC (Hartree-Fock) should give the correct leading order behavior. For the ED calculations, we use pyED [114] from the TRIQS framework [115]. The results for all three methods as a function of $U$ and $J$ are plotted in Fig. 8. Both TPSC3 and TPSC5 show indeed the correct leading order behavior for all TPEVs, irrespective of the ratio of $U$ and $J$. Furthermore, we observe that TPSC5 performs significantly worse when $J$ becomes unphysically large, while TPSC3 stays much closer to the ED result. The TPEV do match well between TPSC3 and ED throughout the whole range of examined parameters. In general, the deviation is strongest in the pair-hopping component, in which the error monotonously increases with $J$. We stress again that since we stay in the weak-coupling regime, these benchmarks have no direct implications for applications in the strongly correlated regime.

To complement the results presented in Fig. 2, we show the same comparison of ED and the TPSC variants at lower filling (upper panel) and with additional inter-orbital kinetic coupling introduced in Fig. 7. The results do qualitatively agree with Fig. 2. As expected, the TPEV are closer to ED at lower filling.



Figure 8: Comparison between TPSC3 (blue), TPSC5 (yellow) and ED (red) for the model defined in Eq. (E.1) at $\beta = 0.5$ on the top and $\beta = 2$ on the bottom, for different $U$ and $J$ combinations. We include the zeroth order self-energy in both TPSC variants.

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
