# Peer review of "Multi-orbital two-particle self-consistent approach -- strengths and limitations"

_SciPost Physics, doi:SciPost Phys. 19, 026 (2025)_

## Round 1 · Referee Report · Anonymous (Referee 1) · 2024-11-25

Report

In this manuscript, the authors present a revised formulation of the multi-orbital Two-Particle-Self-Consistent (TPSC) method, addressing inconsistencies in a previous attempt. They compare the numerical results of the new formulation with those from the original approach and other many-body methods, including DMFT, ED, and D-TRILEX.

While the manuscript may warrant publication in some form, there are several open issues that, in my opinion, need to be addressed before I can recommend it for publication.

Following the Editor advice, I have revised the v2 version that can be found on this link https://arxiv.org/abs/2410.00962.

1.) The authors propose two different formulations of the multi-orbital TPSC method, referred to as TPSC3 and TPSC5. I am curious why this distinction arises. If one starts with the assumption that the two-particle Green's function is constructed as its disconnected part multiplied by a time-independent constant—chosen to reproduce the exact same time and position average—shouldn’t this lead exclusively to TPSC5? If this is not the case, could the authors clarify the source of this ambiguity? Additionally, a diagrammatic representation of the two formulations in terms of vertices could be helpful for less-expert readers to better visualise and distinguish the different approaches.

2.) Another point of concern is that the authors compare the TPSC results on the square lattice with methods that are not exact in two dimensions. Could the authors elaborate on the specific limitations of DMFT and D-TRILEX in capturing physics in two-dimensional systems? Additionally, could quantum corrections in two dimensions further reduce the double occupancy values? Providing clarification on these aspects would strengthen the context of the comparisons and the interpretation of the results.

3.) An interesting point raised by the authors concerns the limitations of the theory in the half-filled SU(4)-symmetric case at J =0 and with zero inter-orbital hopping. This is a peculiar case where the TPSC formulation clearly breaks down, and the authors acknowledge and discuss this, which is an important critical observation. However, this breakdown occurs in a very specific limit. I believe that for the SU(4) case at different integer fillings (i.e., n=1,3), variations in the chemical potential relative to the half-filled case could lead to differences from the single-band model. How might these differences affect the comparison between single-orbital and multi-orbital TPSC?

4) One of the strengths of TPSC is its ability to compute a non-local and frequency-dependent self-energy with reasonable computational cost. However, the authors do not discuss how the new formulations might alter this quantity compared to the previous approach. I believe the authors should address this by calculating the self-energy numerically and comparing it with other many-body methods capable of evaluating this quantity such as DMFT and D-TRILEX.

5) I am uncertain whether comparing TPSC against ED provides meaningful quantitative insights for its potential applications. TPSC is intended for systems in the thermodynamic limit within the weak-to-intermediate coupling regime. A dimerized state may emerge in one-dimensional systems or when interaction strengths exceed the range where TPSC is effective. Additionally, a two-site model lacks a bulk structure, meaning there is no distinction between open-boundary (OBC) and periodic-boundary conditions (PBC). Since TPSC is typically applied to models with PBC, using a cluster (e.g. four site with PBC in 2D) that respects these conditions might offer a more appropriate basis for quantitative comparison.

Requested changes

Here there are some minor aspects that the authors could address for improving the presentation of their paper.

(i) Why does Figure 2 appear before Figure 3, despite the first reference to Figure 2 occurring after the first reference to Figure 3 in the text?

(ii) The first three panels on the left in Figure 3 depict the same quantities. I suggest displaying only one panel and noting in the caption that the three averages should be identical.

(iii) In the legend of figure 4 Ref. 88 is indicated. Did the authors mean Ref.90?

Recommendation

Ask for major revision

  • validity: high
  • significance: high
  • originality: good
  • clarity: good
  • formatting: good
  • grammar: good

Author:  Jonas Profe  on 2025-02-20  [id 5237]

(in reply to Report 1 on 2024-11-25)

See attached pdf for a point by point reply of the points raised by the referee. The redlined manuscript is put as a remark to the comments of referee one.

Attachment:

referee_reply_1.pdf

---

## Round 1 · Referee Report · Camille Lahaie (Referee 2) · 2024-11-26

Strengths

a. The two-particle self-consistent approach has proven quite successful in describing the physics of the one-band Hubbard model in two dimensions. It satisfies several exact results. A generalization of this method for general multi-orbital problems is called for. Despite a few previous attempts, the literature has not converged yet towards a unique, benchmarked approach. This paper provides a generalization of the multi-orbital two-particle self-consistent approach, without relying on Dynamical mean filed theory, that it compares with previous works. It constructs 5 different spin-dependent Ansätze and 3 different spin-independent Ansätze that it benchmarks at half-filling.
b. Long derivations that need to be in the literature are presented.
c. They made it possible to close the system of equations for the vertices using sum rules and symmetries.
d. There is an honest declaration of the limitations of the method.
e. The method benchmarked the local vertices with Exact Diagonalisation and Dynamical Mean Field Theory, which are well-known in the field. (See however reservations in the report).

Weaknesses

a. It needs further explanations or reference for some statements.
b. Many calculations remain obscure and even probably incorrect as written (typos?).
c. There is a lack details in the benchmark sections on the correspondence with the model parameters and the values of the D,P and C parameters. This makes the results and conclusions hard to validate.
d. There are some statements/conclusions that are unclear.
e. Some of the notation is unclear or undefined and even inconsistent.

Report

This report is for the original submission of this paper. It was discussed with André-Marie Tremblay.
The two-particle self-consistent approach (TPSC) satisfies several exact results, like sum-rules, conservation laws and Pauli principle. It also satisfies the Mermin-Wagner theorem in two dimensions. Other methods, such as RPA violate several of these exact results and others like the renormalization group are much more difficult to implement and require more computing power. This makes TPSC a method of choice for the Hubbard model. Its generalization to multi-orbital models is an important and non-trivial next step that this paper intends to achieve. It is appropriate in principle for Scipost.
Unfortunately, in its present form, this paper cannot be published. It lacks justifications of some steps, presents benchmarks that are not always relevant, contains many typos and lacks some details that would be necessary for reproducibility. I suggest a partial list of changes and verifications of the derivations to make sure possible mistakes are avoided. After the authors respond to the constructive criticism below and after further review by me or some other referee, I hope publication can be recommended.

Requested changes

N.B. The changes in notation as one progresses through the paper make it difficult to find out what is a typo and what is a mistake. I have rederived most equations and indicated where I have questions, but I had to stop at some point. 1. In eq 1 : The interaction term should be 1/4 if you are assuming that the matrix element is antisymmetric, as is shown in eq 57 of Salmhofer and Honerkamp, that you reference. I personally think one should immediately start from the unsymmetrized form of the matrix element, and keep ½. 2. Also, in Eq.(1), why are you keeping such a general form of the interaction while you are assuming, without saying it explicitly, that the two-body interaction called U here is spin-independent (as follows from Eq.(2)). 3. After eq. 1 : You define the creation (annihilation) operator with position and imaginary time indices, but in eq. 1, all operators are not explicitly imaginary time-dependent since the Hamiltonian, as usual, is imaginary-time independent. Also, you did not define $\tau_1$ as imaginary time. I know you need to define creation (annihilation) operators with all the position, spin, imaginary time and orbital dependences since you need it later. 4. In eq.3, the term C contains $S \cdot S$ which includes $S_z S_z$, so it does not contain only spin flips that are given by $S_x S_x+S_yS_y = (S_+S_- + S_- S_+)/2$. Shouldn’t it be impossible to have same spin for all the creation (annihilation) operators, because then it isn’t really a spin-flip? The name seems inappropriate. 5. Throughout the derivations (eq 4 and after) : Why did the order of indices change? Please keep the same order of numbered indices as in previous equations, otherwise it makes the derivation hard to follow. 6. You also use two different types of notation : an explicit one (e.g. $\Sigma_{o_1 s_1}(r_1, \tau_1)$) and an abbreviated one (e.g $ \Sigma_1$). First, I would define what $\Sigma_1$ stands for. Second: I would suggest that you keep the same notation throughout, at least within the same eq. I do know how long those equations are, so you try to keep them more concise, but the most important thing is to explain when and how you use a short-hand. (For example, in the appendix, you use both at the same time, and when there are deltas, it is not clear whether these are delta functions for all variables, or just for position and time or otherwise.) 7. In eq. 6 you seem to change notation from position and imaginary time dependent creation (annihilation) operators to non-explicitly dependent ones: Is this to make the eqs. simpler and easier to follow? Please comment on that. 8. Going from eqs. 5 to 6 : I am not sure how you handled the switch in operators and the indices of U. Could you elaborate on that? When I try to do it, I switch the two annihilation operators of $o'_{3} s_1$ and $o'{2} s'$, and because they anticommute, only the sign changes. Then I switch $o'_2$ and $o'_3$ for both U and the operators. That leaves me with two identical four-point function but a sum of $U_{o_2' o_3' o_4' o_1}$ and $U_{o_3' o_2' o_1 o_4'}$. Although, because of the symmetry used, they are equivalent like you said and it cancels the 1/2, but I don’t have any $\delta$ function or $(1-\delta)$. I think they should normally come from the model. And, by the way, it would have been easier to start from an interaction matrix element that was not antisymmetrized. 9. Definition of $\bar{\delta} $: “$\bar{\delta}$ is a short hand for $1-\delta$”: In eq. 6 I read this as : $\bar{\delta}{s_1,s_4'} \bar{\delta}} = (1 - \bar{\delta{s_1,s_4'} ) (1- \bar{\delta} )$ : Which cancels all the aligned spin four-point functions, even when the orbitals are different. Thus, no Hund’s coupling of aligned spins on different orbitals. In the benchmark, this is the term on the right in eq 50. Hence, if this above definition is the one you mean, I think it makes the benchmark a lot less accurate, and that this would explain why the case of the Kanamori Hamiltonian works better, because the other terms in the Kanamori model are not neglected by this. Although, when I follow your derivation, I think you mean it in this way $\bar{\delta}{s_1,s_4'} \bar{\delta} )$} \rightarrow (1 - \delta_{s_1,s_4'} \delta_{o_2',o_3', so that it respects the Pauli principle, but this should come from the model directly, as in eq 50. The only zero term is the case where two electrons are destroyed on the same orbital with same spin. Was this your intention? Could you clarify, or add a section in the appendix with the whole derivation in details? 10. Another way to see that the product of delta functions that appear in eq.(7) are incorrect (unless redefined as above) is to look at the numerator of eq.(9). If there is $\bar{\delta}_{o_1,o_4}$ as a prefactor of D in eq.(7), it must be a prefactor for the whole numerator of eq.(9). The numerator of eq.(9) is correct I think, but not the delta functions in eq.(7). 11. In eq. 7 : If we follow your delta, why are there $1- \delta$ for all terms except for the P term? Could you clarify? When I redid this calculation, nothing mathematical would have removed a $1-\delta$ over orbitals. 12. In eq. 7: I think there is a typo : There is a sum over spin s_4', but primed variables already means that they are summed over in your notation. 13. Eqs 9 through 14 : I do realize the ambiguity in the spin or spin-independent Ansätze, I have three interrogations upon this realization: the first one is that the SU(2) Symmetry on the interaction U allowed to remove the spin-dependency of the interaction term, why is it ok to reinsert it? The second one is how can you differentiate the two different interaction terms of eq 50 with a spin independent interaction? The third one is that while trying to recover this Ansätze, I came upon the realization that we have to assume that the Ansätze are all decoupled, for D, C and P what is your insight on that? 14. In eq.8 the global sign for P is incorrect if eq. 7 is correct. 15. Why are the indices of D interchanged in going from Eq.(8) to Eq.(9)? 16. Eq. 15: Please comment on why you did not put any $(r,\tau)$ dependence. 17. Eqs. 15 and 16 : Only label one number for the whole eq. 18. Eqs. 21 and 22, you analyze the functional derivative contribution of the Ansätze, but only for one term of D and one of C. What about the P coefficient? Also, is it the same result in the case where the Ansätze is spin-independent? 19. What assumptions are involved in claiming that the functional derivatives on the D Ansätze coefficients of eq. 15 vanish in eq. 21. 20. Eq. 21 : I think the left-hand side of the eq. needs to have a sum over s_4? Also, I think that normally, there should be a sum of the orbital indice $o_4$ of $\tilde{D}_{o_4, o_1}^{s_4, \sigma}$ , which is not the same index as the one in the Green’s function of the functional derivative. 21. Eqs. 23 and 24: Following your expression for the self-energy in eqs. 15 and 16 and the definition for Vertex in eq. 20, the sign in both eqs. 23 and 24 should be the opposite. 22. In eq. 34 through 38, it took me a while to understand that 1/2 meant either 1 or 2 and not spin half. Could you maybe just mention a clarification? Once I understood, the notation made sense, but it would be easier for a first-time reader if you put a small explanation or example. 23. In Section II.A : You write “We include both the particle-hole symmetrized and the usual version of the Ansätze in our implementation.“ Does usual mean the one you developed above? What is the particle-hole symmetrized Ansätze? Did you take it from another reference? If so, please add the reference and the eq. number, if you symmetrized your Ansätze, please show the final eq. of those. 24. In Section II.A : You write “Apart from these differences, the sum rules utilized in Ref. [87] and Ref. [88] are a subset of the sum rules employed here, and we checked that by constraining the eqs. we can reproduce the previous results.”. Could you add an appendix on those verifications, what are the constraints that you added? I obtained the same sum rules as you did, but not the ones in the reference you mention. 25. After eq 50 : You talk about how U=U', but you did not mention U' before, please define it or refer to its definition. 26. Between eqs. 50 and 51, about half-filling : What is the definition of your half-filling? One electron per orbital, or per site? Also, could you explain a little more on how you obtain eq. 52 by explaining the values of $n_{o_i,o_j}^{s_i,s_j}$ in the half-filling definition? It could be added in an appendix. 27. Eq 49 : It looks like the numbered orbitals are mixed up, because the right-hand side of the eq. does not close a bubble if we follow your definitions in the appendix. 28. Eq. 57 has two $c^\dagger$ in the hopping term. Is this intended? Is it Hermitian? 29. Section III : You did not mention what was the size of the real-space and reciprocal space used to do TPSC calculations. In figure 3, you seem to be doing TPSC for two site, but even in the single-orbital case, TPSC is not valid below a 64 site x 64 site, and normally it is more used in 128x128 or 256x256, and even 512x512 in the low temperatures regimes. Could you please mention what is the system size that you are using? If it is different for different benchmarks, please mention each time. Also, if it is a 2-site only, then it is expected that TPSC fails, yes. 30. Section III: Benchmarks and models : It could be a good thing to recall the different interaction terms of eq. 7 and give the values you used in the calculations. For both eqs. 56 and 50. Also, in the case of spin-independent Ansätze, how did you differentiate the (U-2J) and (U-3J) in eq. 56? 31. Section III.A, page 8, second paragraph : I think you should add a reference on the results of TPSC for the single orbital model. 32. Section III.A, page 9, first paragraph : The sentence : “As an illustration, let us considering a dimer at half-filling with a Hubbard- Kanamori interaction.” I think you mean “[…] let us consider a dimer […]”. 33. Section III.A, page 9, first paragraph : About the illustration of a dimer at half-filling with a Hubbard Kanamori interaction, you mean the $H_{int}$ of equation 56? If so, I think it would be best if you moved that to section III.B. where eq. 56 is already introduced. If not, could you add the details (eq. number): which interactions exactly you are referencing to? Also, could you show what are the different TPEVs? Which one of the TPEVs is represented by the Hartree-Fock decoupling? TPSC is not expected to be valid in the atomic limit, t=0. So your comment about the fact that the basics ideas of TPSC breaks down is correct, but this was also expected, you could add a reference, from many TPSC papers. 34. In section III.A : after eq 50 : you write “This interaction in combination with the kinetic term leads to a vanishing Σ0” what is your $\Sigma^0$? The non-interacting self-energy? The non-interacting self-energy should be zero, yes, but because the non-interacting green’s function is the one that solves the non-interacting Hamiltonian, which solves exactly, thus there is no self-energy in its expression. Please explain what is your $\Sigma^0$. 35. In section III.B., in the second paragraph, you explain why the higher values of J works better for TPSC while looking at the different groups of spins. You talk about the eigenvalues, but there is not further explanation on how to obtain those eigenvalues. I guess it would come from the diagonalization of the Interaction part only of the Kanamori Hamiltonian, but it would be nice to have more description about that, in an appendix perhaps, or maybe you could add a reference where it is well explained? 36. In section III.B. third paragraph, you write “ This also indicates that this specific part of TPSC performs better at lower temperatures (since it is basically a ground state targeting approach) - however, the local and static approximation of the vertex becomes more and more inappropriate at lower temperatures which is why TPSC typically fails at low temperatures.” TPSC is not “basically a ground state targeting approach”. It is actually far from it. It is not valid in 2D deep in the renormalized classical regime, which is a region of the phase diagram in temperature and density that starts at finite temperature and goes to zero temperature. Also, each part of the sentence separated by “-” are two contradicting statements. In general, I would say that TPSC can reach lower values of temperature than DMFT, but one needs to be careful in the grid definition, because the correlation length should be smaller than the lattice size to capture long wavelenght correlations. A lot of those explanations can be found in [Gauvin, Phys Rev B 108, 7 (2023). DOI : 10.1103/PhysRevB108.075144].

  1. Appendix A, eq. A23 : Normally, the two derivatives w.r.t. $\tau_a$ and $\tau_b$ for $G(\tau_a, \tau_b) $give a different order for $\Sigma G$ : one is $\Sigma G$ and the other is $G \Sigma$.
  2. In eq. A14, what does $S^0$ superscript mean?
  3. The definition of susceptibilities in eq. (A11) does not lead to the matrix structure in eq. 25.

Recommendation

Ask for major revision

  • validity: low
  • significance: high
  • originality: high
  • clarity: low
  • formatting: good
  • grammar: reasonable

Author:  Jonas Profe  on 2025-02-20  [id 5238]

(in reply to Report 2 by Camille Lahaie on 2024-11-26)

See attached redlined manuscript for easier tracking of changes made during the last revision.

Attachment:

redline.pdf

Author:  Jonas Profe  on 2025-02-20  [id 5236]

(in reply to Report 2 by Camille Lahaie on 2024-11-26)

See attached pdfs for point by point reply and redlined manuscript (note that the Equations are not redlined but we reworked every single one to have a more streamlined notation)

Attachment:

referee_reply_2.pdf

---

## Round 4 · Referee Report · Camille Lahaie (Referee 2) · 2025-3-10

Strengths

  1. This paper shows how to solve a multi-orbital Hubbard system with SU(2) symmetry with TPSC. This is a great addition to the domain as TPSC has been known to show physical features unaccessible to Dynamical mean-field calculations or Monte-Carlo ones.
  2. The derivations are thorough and well explained, and the author added many details in Appendices, which are all really pertinent.
  3. The analysis of the limits of the Hartree-Fock decoupling is really interesting and gives insights to the limitations of TPSC in its pure form.
  4. The method is benchmarked against a previous formulation of multi-orbital TPSC, DMFT and D-TRILEX

Weaknesses

  1. Some of the claims still need further explanation and/or references.
  2. There are still some mathematical errors and/or typos with small inconsistencies, but it is almost all resolved from the first submission.

Report

I have followed the redprint for my comments. It was discussed with André-Marie Tremblay.
The two-particle self-consistent approach (TPSC) is an appealing method to study strongly correlated systems. It satisfies many exact results and even though multi-orbital formulations were already done before, this generalization in the SU(2) symmetry which is presented in this paper is non trivial and will be a great addition to the field.
Unfortunately again, in the present form, this paper cannot be published. It needs small corrections of mathematical issues. Furthermore, there is a section in which the discussion needs either more results shown, a physical explanation of some references to back up some statements.

After the authors respond to the constructive criticism below and after further review by me or some other referee, I hope publication can be recommended.

Requested changes

  1. After the Eq. (16), the following definition is shown : $n^{s_1 s_2}{o_1 o_2} = c^{\dagger})$} (\tau, \mathbf{r}) c_{o_1 s_1} (\tau, \mathbf{r. First of all, I think the previous introduced notation should be kept here $\boldsymbol{\tau}=(\mathbf {r},\tau)$ , for consistency. Second, shouldn’t there also be a $\boldsymbol{\tau}$ dependance to $n^{s_1 s_2}_{o_1 o_2}$? Like such : $n^{s_1 s_2}{o_1 o_2}(\boldsymbol{\tau}) = c^{\dagger})$} (\boldsymbol{\tau}) c_{o_1 s_1} (\boldsymbol{\tau ?

  2. For equation 8, but also for all of the other equations where the sum are redundant. I have to stress that equations written as is, with two sums over one index, is not mathematically valid: at least not for the equation that is written. There are many ways to keep the equation mathematically valid and explicitly show the sum: either to remove the prime over the indices that are linked with the summation operator. eg. $\sum_{s_4} c_{s_4} c_{s_1}$. To make an even greater distinction, one could use a different notation for the explicitly summed indices: e.g. $\sum_{\alpha} c_{\alpha} c_{s_1}$.

  3. In the same vein as the previous point, the cases of sums shown explicitly to take into account excluded orbitals, e.g. eq. (47), the $1-\delta$ could be used: keeping the indices primed and adding $1-\delta$ would mean exactly what is written in the text. e.g. $ (1-\delta_{o_1 o_2}) c_{o_1} $.

  4. In Section 3.2, there is a discussion about how the approximation of local and static vertices break down as one lowers temperature. Why? First of all, the results shown in the paper are almost all at the same temperature , which is mostly a high temperature. There are no results comparing to DMFT vertices as the temperature goes down. In the papers mentioned, I have not seen any mention of locality of vertices breaking down. I want to stress here that I am not fully against that possibility, but it seems to me that this is a claim that needs backup or a physical explanation which I have not found here. Secondly, single-orbital TPSC does not fail at low temperature, except in the renormalized classical regime. And even in the renormalized classical regime with static and local vertices for doubly self-consistent TPSC, there was great correspondance with benchmarks. [Vilk, Y. M., et al. PRB 110, no. 12 (2024): 125154. https://doi.org/10.1103/PhysRevB.110.125154.]

Recommendation

Ask for minor revision

  • validity: good
  • significance: top
  • originality: top
  • clarity: top
  • formatting: excellent
  • grammar: excellent

Author:  Jonas Profe  on 2025-04-17  [id 5381]

(in reply to Report 2 by Camille Lahaie on 2025-03-10)

See attached redlined manuscript for easier tracking of changes made during the last revision.

Attachment:

redline.pdf

Author:  Jonas Profe  on 2025-04-17  [id 5380]

(in reply to Report 2 by Camille Lahaie on 2025-03-10)

See attached pdf for a point by point reply of the points raised by the referee.

Attachment:

reply_referee_2.pdf

---

## Round 4 · Referee Report · Anonymous (Referee 1) · 2025-3-10

Report

The new version of the manuscript is an improvement when compared to its previous one however I still have some doubts about one of the main aspect of the paper and therefore I am not convinced to recommend this paper for publication yet.

1.) From what I understand, the authors argue that the source of ambiguity arises from the structure of the equation of motion, specifically whether the sum over the internal spin index is incorporated into the ansatz definition or not. However, in TPSC, the ansatz is typically chosen to reproduce the correct static and local limits of the two-particle Green’s functions. This can be achieved by defining G(2) without directly relying on the equation of motion. Once this quantity is fixed at the level of G(2), it is then used in the equation of motion.
My question is: would this ambiguity persist if one were to adopt this protocol for fixing the ansatz?

2.) I would honestly suggest the authors to move Figure 2 into the appendix.

3.) The authors considered my advice to perform additional calculation partially. While I appreciate their effort in doing so I think that it would be informative to compare the self-energy computed using the different methods fixing two or more k-points as a function of frequency.

Recommendation

Ask for major revision

  • validity: -
  • significance: -
  • originality: -
  • clarity: -
  • formatting: -
  • grammar: -

Author:  Jonas Profe  on 2025-04-17  [id 5379]

(in reply to Report 1 on 2025-03-10)

See attached pdf for a point by point reply of the points raised by the referee.

Attachment:

reply_referee_1.pdf

---

## Round 4 · Author Response

We thank the referees for their helpful and constructive comments.

---

## Round 4 · List of Changes

Reworked the whole manuscript and all equations such that the notation is unified. Added three Appendices (App. B,C,D) and shifted the ED results from the main text to Appendix E. See redlined manuscipt provided in the reply to the referees for a detailed list of changes.

---

## Round 6 · Referee Report · Camille Lahaie (Referee 2) · 2025-5-5

Strengths

This paper shows how to solve a multi-orbital Hubbard system with SU(2) symmetry with TPSC. This is a great addition to the domain as TPSC has been known to show physical features inaccessible to Dynamical mean-field calculations or Monte-Carlo ones.

The derivations are thorough and well explained, and the author added many details in Appendices, which are all really pertinent.

The analysis of the limits of the Hartree-Fock decoupling is really interesting and gives insights to the limitations of TPSC in its pure form.

The method is benchmarked against a previous formulation of multi-orbital TPSC, DMFT and D-TRILEX

Weaknesses

The train of thoughts in the benchmark section can be hard to follow, however, I think this is not an issue that can easily be resolved, and the paper has still proven to be consistent.

There are still some typos.

Report

I thank the authors for the first two replies of the first and second submission.

I have followed the redprint for my comments. It was discussed with André-Marie Tremblay.

The two-particle self-consistent approach (TPSC) is an appealing method to study strongly correlated systems. It satisfies many exact results and even though multi-orbital formulations were already done before, this generalisation in the SU(2) symmetry which is presented in this paper is non trivial and will be a great addition to the field.

I think publication can be recommended only after the authors modify the small changes and answer some questions.

Requested changes

1- (Question) In Sec 3: After eqs 48, 49 and 50, that show the values of the 3 vertices of TPSC as a function of U and J of the Kanamori model. We see that the right-hand side of eqs 49 and 50 are equal, does that also mean automatically that P=C? I would expect that this is not the case for TPSC5, right?

2- In Eq 51: Shouldn’t there be a sum over $k$ and $\nu$? Or maybe we used Einstein’s notation and I missed the comment? But I also think there should be a normalisation term here?

3- Eq 52: One U should be U'? It is mentioned below that equation that U=U', so perhaps U' should appear in that equation. Or maybe looking at the Kanamori model, they should not be separated as such and just mentioning their properties (intra- or inter- orbital) is enough. I am just asking to make sure it is also clear for readers.

4- One suggestion, it is up for the authors, but I think readers would be more inclined to follow the discussion if the comparisons to DMFT were done before the isolated case of J=0 discussion, which shows and explains why TPSC fails at low J and not so much at higher J. (So Sec 3.2 before Sec 3.1) But I do understand in a way why it was done the way it is right now. So I leave it to the authors to decide.

5- After Eq 59: I think there is a typo, the sentence starts with “This we can …”.

6- In the caption of the plots of Figure 3 : I think the number of the Reference was not updated, because it is not the same as in the citation of the same figure.

7- Last paragraph of Sec. 3.3.2 : There is no direct reference to the figure that shows susceptibilities (Fig 6, I think) , even though there is a discussion about it. This seems to be just an oversight.

8- Figure 2b) : From what type of calculation do those results come from? Exact diagonalization, DMFT?

9- In Section 3.2 : There is an explanation of the limitation of TPSC at low temperature coming from the Hartree-Fock decoupling, but then in Appendix E, it is said that Hartree Fock is more valid at low temperature, this looks like a contradiction. After a couple of re-reading, I have come to understand that the limitation at low temperature for TPSC is due to the importance of the non-local and non-static corrections on the vertices, but doesn’t that also apply to Hartree-Fock?

Recommendation

Publish (meets expectations and criteria for this Journal)

  • validity: high
  • significance: top
  • originality: top
  • clarity: good
  • formatting: perfect
  • grammar: excellent

Author:  Jonas Profe  on 2025-06-18  [id 5580]

(in reply to Report 1 by Camille Lahaie on 2025-05-05)

Here is the redlined manuscript for easier tracking of changes.

Attachment:

redline.pdf

---

## Round 6 · Referee Report · Anonymous (Referee 1) · 2025-5-19

Report

The authors have satisfactorily addressed my comments and strengthened the manuscript by adding additional calculations. The revised version now meets the criteria for publication in SciPost.

Recommendation

Publish (meets expectations and criteria for this Journal)

  • validity: high
  • significance: high
  • originality: good
  • clarity: high
  • formatting: excellent
  • grammar: excellent

Author:  Jonas Profe  on 2025-06-18  [id 5579]

(in reply to Report 2 on 2025-05-19)
Category:
answer to question

See attached pdf for a point by point reply to the referee.

Attachment:

reply_referee_2.pdf

---

## Round 6 · List of Changes

See redlined manuscript in comment to referee 1

---

## Editorial Decision

published